# Predictive neural representations of naturalistic dynamic input

**Ingmar E. J. de Vries** [1,2] ✉ **& Moritz F. Wurm** [1]

Adaptive behavior such as social interaction requires our brain to predict unfolding external dynamics. While theories assume such dynamic prediction, empirical evidence is limited to static snapshots and indirect consequences of predictions. We present a dynamic extension to representational similarity analysis that uses temporally variable models to capture neural representations of unfolding events. We applied this approach to source-reconstructed magnetoencephalography (MEG) data of healthy human subjects and demonstrate both lagged and predictive neural representations of observed actions. Predictive representations exhibit a hierarchical pattern, such that high-level abstract stimulus features are predicted earlier in time, while low-level visual features are predicted closer in time to the actual sensory input. By quantifying the temporal forecast window of the brain, this approach allows investigating predictive processing of our dynamic world. It can be applied to other naturalistic stimuli (e.g., film, soundscapes, music, motor planning/ execution, social interaction) and any biosignal with high temporal resolution.

In dynamic environments such as social interaction (e.g., traffic or sports), our brain is faced with a continuous stream of changing sensory input. Effective and prompt interaction with such an environment requires our brain to continuously generate predictions of unfolding external dynamics. Although theories assume such dynamic prediction[1–3], empirical research has focused on static snapshots and indirect consequences of predictions, often using simple static stimuli of which predictability is manipulated[4–8]. It is unclear how these findings generalize to our complex dynamic world. For example, to plan appropriate motor commands for catching a ball, the brain needs to estimate its future trajectory based on the opponent's intentions, the throwing movement, and the ball's initial trajectory. While this clearly demonstrates the brain's ability to predict, previous analytic frameworks preclude investigating what the brain predicts at which point in time, particularly across hierarchical levels of stimulus complexity in such naturalistic dynamic environments, thus leaving several fundamental questions unanswered. For instance, while observed actions have been shown to be represented at distinct, hierarchically organized levels[9,10], it is unclear at what hierarchical level the brain makes predictions, what the causal relationship is amongst predictions at different levels, and how these predictions temporally relate to each other. While predictive processing theories suggest that higher-level predictions act on, and therefore must precede lower-level predictions[11–13], low-level prediction of simple moving objects is also possible without high-level prediction[14], suggesting (partly) independent prediction streams. Another unsettled theoretical debate regards prediction errors. According to predictive coding theory, post-stimulus neural representations should solely reflect the difference between prediction and sensory input (i.e., unpredicted input or prediction errors)[15]. As such, accurate predictions effectively silence (or explain away) all sensory input, and no post-stimulus representation is expected. In contrast, related theories such as adaptive resonance[16], or Bayesian inference without predictive coding[17], hypothesize that top-down prediction and sensory input combine to provide the most accurate possible post-stimulus representation. Here we present a framework that allows investigating these unresolved questions by quantifying how representations in the brain temporally relate to (i.e., follow or precede) actual events across hierarchical levels of complexity in naturalistic dynamic stimuli.

This framework builds on representational similarity analysis (RSA), a powerful approach to investigating neural representations[18], which quantifies the similarity between neural and model representations.

[1]Centre for Mind/Brain Sciences (CIMeC), University of Trento, 38068 Rovereto, Italy. [2]Donders Institute, Radboud University, 6525 EN Nijmegen, The Netherlands. ✉ e-mail: i.e.j.de.vries@gmail.com

RSA has been mainly used to investigate neural representations of static stimuli (e.g., pictures of objects), using static models (e.g., for shape or category)[19]. To investigate representational dynamics of temporally changing stimuli, we developed a dynamic extension to RSA that uses temporally variable models to capture neural representations of unfolding events (Fig. 1). In dynamic RSA, the similarity between neural and model representations is computed at all neural-by-model time points. Hereby, dRSA allows investigating the match between a model at a given time point and the neural representation at the same or different time points (i.e., earlier, or later). In the case of pure bottom-up processing, one expects a lag between a time point-specific visual model state and the best-matching neural representation (i.e., the time needed for information to pass from the retina to e.g., V1). In contrast, a negative lag should be observed for predictive neural representations, in which case neural representational content predicts the future model state. However, note that while a negative dRSA peak latency clearly evidences prediction, a positive latency precludes inferring that a representation is solely activated in a bottom-up manner. That is, even post-stimulus representations are likely to be modulated by top-down expectations[2], for instance by sharpening the representation[20], or by shortening the processing latency[8,21]. Further note that as moment-by-moment stimulus predictability is determined by stimulus-specific feature trajectories and event boundaries, the resulting dRSA peak reflects the average latency at which a feature is represented most strongly for these stimuli. It does not mean this feature is exclusively represented at that exact latency.

Here we applied dRSA to source-reconstructed magnetoencephalography (MEG) data of healthy human subjects observing dance videos that were modeled at various levels of abstraction across the visual processing hierarchy, from low-level visual (i.e., pixelwise grayscale) to higher-level, perceptually more invariant (i.e., 3-dimensional view-dependent and -invariant body posture and motion), as to capture a comprehensive characterization of the dance sequences. Attention to the unfolding actions was confirmed by high performance on occasional test trials, in which subjects were tested on the dancer's motion, whereas eye fixation was ensured by subjects' successful detection of an occasional subtle color change of the fixation cross displayed in the center (see subsection "Experimental design" in the "Methods" section). Using dynamic RSA, we reveal the peak latency and spread of both lagged and predictive neural representations of naturalistic dynamic input, at several hierarchical levels of processing. By quantifying the temporal forecast window, dynamic RSA represents a methodological advancement for studying when our brain represents and predicts the dynamics of the world.

## Results and discussion
### Hierarchical motion prediction

Dynamic RSA revealed distinct types of dynamic representations with unique characteristics (Fig. 2a): Most importantly, body motion was represented in a broad temporal window preceding the actual input by ~500 ms for view-invariant body motion, and ~200 ms for view-dependent body motion (Fig. 2a and Table 1), indicating that these neural representations predicted future body motion. A computer vision model capturing low-level motion as optical flow vector direction at each pixel was represented predictively at ~110 ms. Predictive representations were present in all regions of the action observation network (AON), albeit strongest in visual areas, which was confirmed by a whole-brain analysis (Fig. 2b). While several previous studies revealed the consequence of motion prediction in early visual cortex[5,22], our results provide a first characterization of how neural prediction of future motion in naturalistic stimuli continuously unfolds at several timescales along the visual processing hierarchy. Specifically, peak latencies of predictive motion representations reflected the order along the processing hierarchy, such that view-invariant body motion was predicted earliest in time, followed by view-dependent body motion and optical flow vector direction (Table 1; main effect of the model on retrieved jackknifed peak latencies[23,24] in a post-hoc ANOVA with factors model and ROI; $F(2) = 19.9$, $p = 8.3e-7$, $\eta_p^2 = 0.49$, 95% CI of condition contrasts = $-0.092–0.240$, $0.229–0.562$, and $0.155–0.488$ for optical

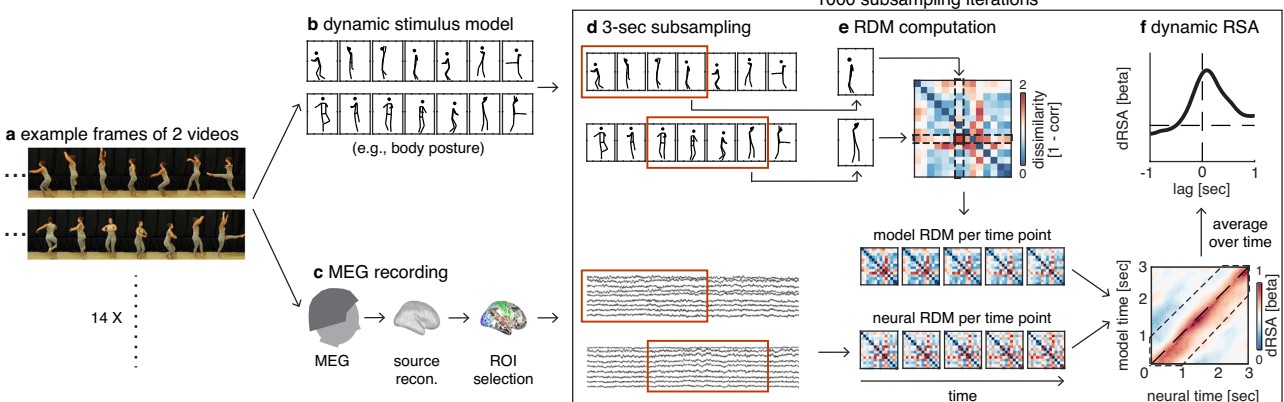

**Fig. 1 | Dynamic representational similarity analysis (dRSA). a** Subjects observed ~36 repetitions of 14 unique 5-sec dancing videos (Table 2 and Fig. 4a) during MEG. **b** Stimuli were characterized at different levels using dynamic stimulus models (e.g., body posture; see Fig. 5 for all models). **c** Individual-subject source-reconstructed MEG signals within regions of interest (ROI; see Fig. 6 for ROI definitions) were used as features for subsequent steps. **d** Temporal subsampling and realignment were used to attenuate idiosyncratic temporal heterogeneity in dRSA results caused by arbitrary pairwise alignment specific to these 14 stimuli (see subsection "Temporal subsampling" in the "Methods" section). In short, across 1000 iterations, a 3-s segment was randomly extracted from each of the 14 5-sec stimuli (orange box), after which these new 14 subsampled 3-s segments were realigned. Crucially, while a different random 3-s window was selected for each of the 14 stimuli, for a given stimulus the temporal alignment between neural signal and models remained intact. **e** Subsequently, on each iteration, neural and model representational dissimilarity matrices (RDM) were created at each time point in the realigned 3-s segments, based on pairwise dissimilarity in neural responses to the 14 stimuli and pairwise dissimilarity in stimulus feature models, respectively (here shown for 5-time points). **f** Last, similarity between neural and model RDMs was computed for each neural-by-model time point (lower panel), using regression weights to test a specific model RDM, while regressing out other covarying model RDMs. This approach was validated through simulations (see subsection Simulations and Fig. 8). Last, the 2-dimensional dRSA matrix was averaged along the diagonal to create a lag-plot (i.e., the lag between neural and model RDM; upper panel), in which peaks to the right or left of the vertical zero-lag midline reflect lagged or predictive neural representations, respectively. These dRSA lag plots are computed within each subsampling iteration, after which they are averaged over iterations separately for each subject, ROI, and model, and statistically tested.

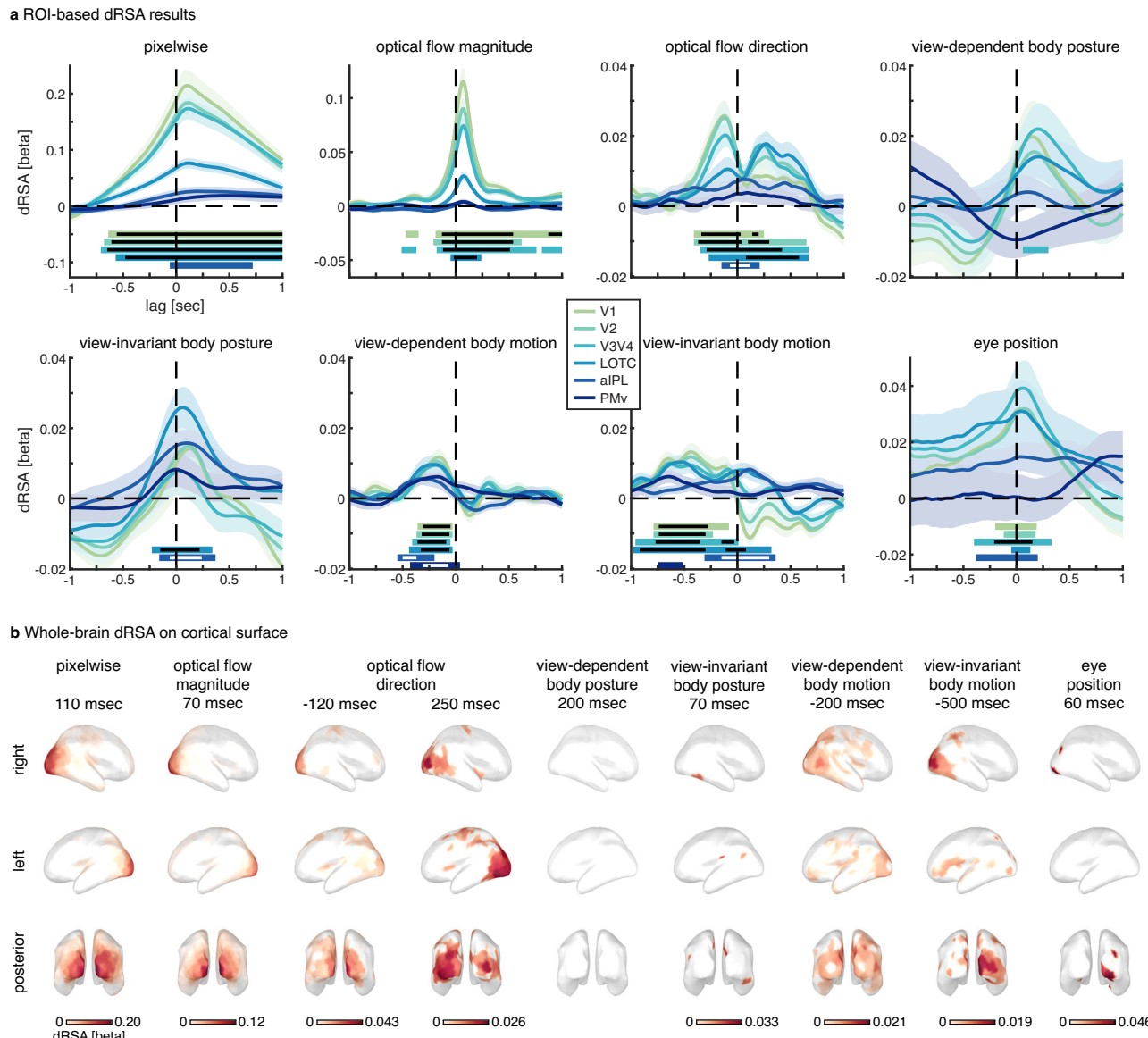

**Fig. 2 | Dynamic representational similarity analysis (dRSA) results. a** Region of interest (ROI)-based analysis, with dRSA regression weights illustrated as lag plots (see Supplementary Fig. 1 for 2D dRSA plots). Light-to-dark colors reflect posterior-to-anterior ROIs (see subsection Source reconstruction and ROI selection). Stimulus-feature models are plotted in separate subplots, with pixelwise capturing low-level visual information, optical flow magnitude and direction capturing low-level visual motion in each of the pixels, and body posture and motion models capturing 3D position and motion of 13 kinematic markers placed on the dancer (see subsection Stimulus models). Additionally, while subjects were instructed to keep fixation, an eye-position-based RDM was included to control for any remaining oculomotor activity (e.g., micro-saccades). Lines and shaded areas indicate subject average and SEM, respectively, with $n = 22$ independent human subjects. Source data are provided as a Source Data file. Horizontal bars indicate beta weights significantly larger than zero (one-sided t-test with $p < 0.01$ for each time sample), corrected for multiple comparisons across time using cluster-based permutation testing ($p < 0.05$ for cluster-size), with colors matching the respective ROI line plot. Black or white horizontal bars inside colored bars indicate the same but with significance at a stricter statistical threshold (i.e., $p < 0.001$ for single sample tests and $p < 0.01$ for cluster-size). **b** Subject average whole brain dRSA on the cortical surface (FDR corrected at $p < 0.05$), with $n = 22$ independent human subjects. Source data are provided as a Source Data file. Peak timings were selected per model based on the ROI analysis illustrated in (**a**). See Fig. 6 for a comparison with the ROIs used in (**a**). LOTC = lateral occipitotemporal cortex, aIPL = anterior inferior parietal lobe, PMv = ventral premotor cortex.

flow direction vs. view-dependent motion, optical flow direction vs. view-invariant motion, and view-dependent vs. view-invariant motion, respectively). Note that this does not evidence a strict serial cascade of prediction, such that low-level prediction is only enabled by earlier higher-level prediction. Instead, some low-level prediction likely remains even without any high-level prediction, as is apparent from the exclusively low-level prediction of simple motion stimuli that do not have a complex naturalistic hierarchical structure[6,14]. The three levels of prediction might therefore partly reflect independent streams. However, the temporal order of predictive representations observed here is in line with hierarchical Bayesian brain theories such as predictive coding that postulate that higher-level predictions act on, and therefore must precede lower-level predictions[11,12,25]. Furthermore, such hierarchical prediction is also suggested by empirical findings from different sensory modalities[26–29] and complex contexts such as social perception and action observation[30,31]. In complex naturalistic stimuli as utilized here one would similarly expect a causal relationship amongst the partly independent prediction streams, such that high-level prediction affects low-level prediction. It may be that predictions originating at a higher, perceptually invariant level, as best captured by

**Table 1 | Peak latencies**

| | Pixelwise | Optical flow magnitude | Optical flow direction predictive | Optical flow direction lagged | View-dependent body posture | View-invariant body posture | View-dependent body motion | View-invariant body posture |
|---|---|---|---|---|---|---|---|---|
| V1 | 100 | 70 | −120 | 270 | – | – | −160 | −440 |
| V2 | 110 | 70 | −120 | 270 | – | – | −180 | −440 |
| V3 + V4 | 110 | 70 | −110 | 230 | 200 | – | −190 | −640 |
| LOTC | 110 | 70 | −90 | 270 | – | 60 | −200 | −520 |
| aIPL | 200 | – | – | – | – | 100 | −320 | – |
| PMv | – | – | – | – | – | – | −180 | −650 |

Peak latencies in msec of the subject-average (*n* = 22 independent human subjects) dRSA curves are displayed in Fig. 2. Note that as we observed both a predictive (negative) and lagged (positive) peak for optical flow direction (Fig. 2a), we computed peak latency separately for negative and positive peaks. Results are shown only for ROI-model combinations with a significant main dRSA result (Fig. 2). Note that because peak latencies of the dRSA curves are difficult to estimate on a single-subject level, we performed statistics on the retrieved jackknifed peak latencies[23, 24] (see the subsection "Peak latency" under the "Methods" section).
*LOTC* lateral occipitotemporal cortex, *aIPL* anterior inferior parietal lobe, *PMv* ventral premotor cortex.

view-invariant body motion, subsequently modulate more concrete predictions, to which eventually the sensory input can be compared. Such causal relationship between different levels of prediction in naturalistic dynamic stimuli remains to be demonstrated in future research, for instance by selectively perturbing high-level stimulus predictability. Two expectations would be a shortening of the temporal forecast window for lower-level features (i.e., a less predictive dRSA peak), and a decrease in the strength of prediction.

Interestingly, for low-level motion (i.e., optical flow direction) we did not only observe a predictive, but also a lagged post-stimulus representation. According to predictive processing theories, post-stimulus representations should reflect the difference between prediction and sensory input (i.e., unpredicted input or prediction errors)[11,12,15,32,33]. As such, it may be surprising that such post-stimulus representation is absent for high-level (view-dependent and -invariant) body motion. However, predictive processing theories hypothesize that accurate predictions effectively silence (or explain away) all sensory input[11,12,15,32,33]. Albeit speculative, it may thus be that due to the highly predictable nature of smooth biological motion as in the ballet stimuli used here, motion information is effectively silenced after the initial comparison at the lowest level, thus explaining the absence of lagged representation of high-level motion models. Such efficient silencing of redundant stimulus information might be particularly important in dynamic stimuli, as new potentially relevant input is continuously incoming. However, if accurate prediction has indeed silenced all sensory input in terms of high-level motion models, it is unclear why this does not hold for the posture models that clearly have lagged post-stimulus representations (see next section). Additionally, even for high-level motion information perfect prediction seems unlikely, and it is worth considering why dRSA might fail to pick up any remaining unpredicted information for high-level motion models. Note that dRSA reflects an average representation across stimulus time, stimuli, and subjects, rather than a single exclusive representation (latency). The little remaining unpredicted information may therefore simply be too variable in latency to be picked up in the average representation. Also, the signal-to-noise ratio (SNR) might generally be lower for bottom-up representations, as the visual cortex receives a much denser network of feedback relative to feedforward projections[33,34]. In any case, these results do not provide conclusive evidence for predictive processing/coding theory (i.e., that accurate predictions silence all sensory input[33]) but leave the door open for related theories that do hypothesize lagged post-stimulus representations even after accurate predictions such as adaptive resonance[16], or Bayesian inference without predictive coding[17]. Future research using dRSA should selectively perturb stimulus predictability at different hierarchical levels as described above to arbitrate between these theories.

## Prediction modulates body posture representation

We observed lagged representations of view-dependent (~200 ms; V3/4) and view-invariant (~60 ms for LOTC and ~100 ms for aIPL) body posture, which might suggest bottom-up activation. However, remember that a positive latency precludes inferring that a representation solely reflects bottom-up activation, as top-down expectations might very well modulate its veridity[20] and processing latency[21]. In fact, we argue that the latency of the view-invariant body posture representation observed here is indeed modulated by top-down prediction. That is, in the case of pure bottom-up processing along the visual hierarchy, one would expect a high-level representation such as view-invariant body posture in LOTC and aIPL to have a larger processing latency compared to low-level representations—which is not what we observed. Instead, the dRSA peak latency for view-invariant body posture in LOTC (~60 ms) is shorter than the peak latency for the view-dependent body posture and pixelwise models (~200 and ~110 ms, respectively; Fig. 2a and Table 1), suggesting that the observed neural representation of view-invariant body posture is at least partly shifted ahead in time by top-down prediction. This interpretation is in line with the observation that prediction shifts the neural representation of the position of simple (apparently) moving objects closer to real-time[8,14], and could explain the subjective experience of perceiving our dynamic environment in real-time rather than lagged[3,14]. While this interpretation might seem intuitive regarding our conscious experience of the world, a real-time position representation fails to explain how we are able to act promptly (e.g., catch a ball), as we would need some representation of the stimulus trajectory to reach the motor cortex clearly well ahead of real-time. Albeit speculative, it may therefore be that the close-to-real-time body posture representation serves our conscious experience, while in contrast, the clearly predictive motion representations serve prompt behavior. Additionally, the exact prediction latencies as identified here, as well as those observed in previous studies, likely depend on stimulus and task characteristics. Future work is needed to identify which factors (e.g., speed, predictability, prior knowledge, attention) determine prediction latencies, and to clarify the apparent difference in representational latency between body posture (i.e., close to real-time) and body motion (i.e., clearly predictive).

## Temporal spread of neural representations

Interestingly, most neural representations were not exclusive to an exact latency, but spread over time, as quantified by comparing the amount of information spread in the neural signal surrounding a dRSA peak with the amount that can be expected based on model autocorrelation alone (i.e., representational spread or RS; Fig. 3a, b). Such spread could be caused by representations themselves being temporally sustained, or by variability in predictive latency across timepoints within a stimulus, across stimuli, or across subjects. One would expect

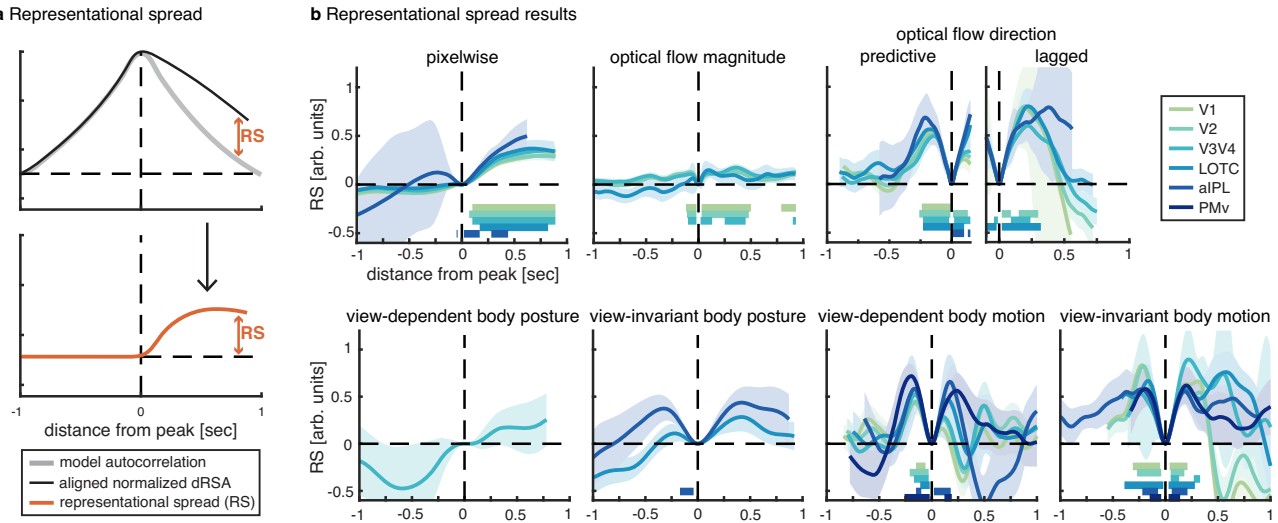

**Fig. 3 | Representational spread.** Results are plotted for ROI-model combinations with a significant dRSA result (Fig. 2). **a** Representational spread (RS), which indicates temporal information spread surrounding the dRSA peak, above and beyond what can be explained by model autocorrelation. To compute RS, we normalized both the observed dRSA curve and the model autocorrelation (i.e., what remains after PCR; Fig. 8b) by their peak value, aligned the curves, and subtracted model autocorrelation from observed curves. RS of 1 and 0 indicate maximally sustained and not sustained representation, respectively. The horizontal axis indicates the temporal distance from the corresponding peak, with distance = 0 coinciding with the peak latency. Positive and negative temporal distances reflect representational spread on the right and left sides of the dRSA peak, respectively. Since the dRSA peak latency determines the temporal window at which the RS can be computed,

different RS lines end at different distances. **b** Representational spread results. Lines and shaded areas indicate jackknifed subject averages and SEM, respectively, with *n* = 22 independent human subjects. Source data are provided as a Source Data file. Same color conventions as Fig. 2a. Horizontal bars indicate representational spread significantly different from zero (two-sided *t*-test with *p* < 0.001 for each time sample), corrected for multiple comparisons across time using cluster-based permutation testing (*p* < 0.01 for cluster-size), with colors matching the respective ROI line plot. Note that as we observed both a predictive and lagged peak for optical flow direction (Fig. 2a), we computed RS separately for each peak. LOTC lateral occipitotemporal cortex, aIPL anterior inferior parietal lobe, PMv ventral premotor cortex.

more spread for higher-level predictions (i.e., of features changing at a longer timescale), and more precise timing (i.e., sharper peak) for low-level models with predictions close in time to the actual sensory input. This is indeed what we observe (Figs. 2a and 3b), i.e., while view-invariant body motion shows the most representational spread on both sides of the peak, this is less so for view-dependent body motion. If in fact representational spread partly reflects a sustained representation, this might be explained by gradually, rather than abruptly, emerging predictions.

For lagged representations, this could indicate that representations were not immediately overwritten by new incoming visual input but remained partly activated. The fact that lagged stimulus information is not immediately overwritten may provide a basis for neural predictions; that is, predicting the future requires the integration of past information across an extended period for extrapolation[3]. As such, the sustained nature of representations might be necessary for creating and updating predictions based on constantly changing dynamic input. An alternative function of sustained representations might be the integration of features to form coherent higher-level representations. As different features have different processing times, an overlapping temporal window of representation is needed for their integration[3]. Future research should investigate how, and which past information is integrated to form predictions, e.g., as shown for simple tone sequences[35], or by investigating the temporal characteristics of predictive representations after feature-specific perturbations. When additionally considering the peak latencies, representations of pixelwise vs. optical flow vector magnitude point towards a dissociation between representations drawing on parvocellular vs. magnocellular pathways, respectively: Pixelwise information was represented later (Table 1; -110 ms) and more sustained (Fig. 3b), whereas optical flow vector magnitude was represented faster (Table 1; -70 ms) and more transient (Fig. 3b). The difference in peak latencies was confirmed by a main effect of model on retrieved jackknifed peak latencies[23,24] in a

post-hoc ANOVA with factors model and ROI (F(1) = 13.3, *p* = 1.5e−3, $\eta_p^2$ = 0.39, 95% CI of condition contrast = 0.02−0.06). Such dissociation is in line with findings suggesting that the parvocellular pathway processes visual information at high spatial resolution with a slow and sustained response pattern, whereas the magnocellular pathway processes motion information with a faster and transient response pattern[36].

## The temporal forecast window of the brain
Taken together, dynamic RSA allows for quantifying the temporal forecast window of the brain. As such, it represents a methodological advancement for studying when our brain represents and predicts the dynamics of the world. It reveals both the peak latency and spread of neural representations of naturalistic dynamic input and does so at several hierarchical levels of processing. The current results go beyond recent evidence for the prediction of simple (apparently) moving stimuli in early brain regions[6,7,14,37], by showing that (1) dRSA reveals prediction at both low and high levels of abstraction in complex naturalistic stimuli and (2) dRSA enables distinguishing these different levels in the same stimulus, thus revealing multiple feature-dependent temporal forecast windows in the brain. The current dataset, reveals hierarchical predictive representations of the future motion of an observed action and an absence of post-stimulus motion representations at higher levels at which sensory input is possibly explained away by accurate prediction. While both these observations support predictive processing/coding theories[11,12,25], future research using dRSA is needed to exclude related theories such as adaptive resonance[16], or Bayesian inference without predictive coding[17].

In principle, dRSA can be applied to any (naturalistic and controlled) dynamic stimulus (e.g., film, soundscape, music, language, motor planning/execution, social interaction) and any biosignal with high temporal resolution (e.g., M/EEG, ECoG, animal electrophysiology,

**Table 2 | Ballet figures per stimulus**

| Stimulus | Ballet figures | | | |
|---|---|---|---|---|
| 1 | Bow left | Arabesque left | Pirouette | Passé back |
| 2 | Bow left | Passé left | Jump right | Pirouette |
| 3 | Bow front | Jump back | Passé front | Arabesque front |
| 4 | Pirouette | Bow front | Passé front | Jump right |
| 5 | Pirouette | Passé right | Bow front | Jump back |
| 6 | Passé right | Bow right | Arabesque right | Pirouette |
| 7 | Passé right | Pirouette | Jump left | Arabesque front |
| 8 | Passé front | Arabesque front | Bow left | Jump right |
| 9 | Jump left | Bow right | Arabesque front | Passé left |
| 10 | Jump right | Pirouette | Passé front | Arabesque right |
| 11 | Jump back | Arabesque front | Pirouette | Bow front |
| 12 | Arabesque left | Pirouette | Jump back | Passé front |
| 13 | Arabesque right | Passé right | Pirouette | Bow front |
| 14 | Arabesque right | Jump left | Bow right | pirouette |

EMG, eye-tracking). Several recent studies investigated the dynamics of neural representations using related approaches. For instance, the temporal generalization approach investigates how neural activity patterns that discriminate between experimental conditions generalize across time[38]. Other approaches used RSA to study top-down processing by computing representational similarity between different brain regions at different time points to estimate the directionality of information flow[39] or used RSA to distinguish words prior to their presentation during naturalistic reading[40]. Importantly, all these approaches have in common that they use static models and/or stimuli. Dynamic RSA extends representational analysis methods to the study of dynamic events, thus paving the way for investigating the neural mechanisms underlying the representation and prediction of information across a wide range of processing levels in naturalistic dynamic scenarios.

## Methods

### Subjects

Twenty-two healthy human volunteers (mean age, 30 ± 7 years) participated in the experiment for monetary compensation. All subjects were right-handed, had a normal or correct-to-normal vision, and were naive with respect to the purpose of the study. As we did not have any hypotheses pertaining to differences in sex or gender, we tried to recruit a balanced dataset, in which 13 out of 22 participants identified themselves as female. Sex or gender information was not collected otherwise. We aimed for our findings to be generalizable to the whole population, and therefore combined all subjects irrespective of sex/gender in the group-level statistical analyses. No sex- or gender-based analyses were performed. All experimental procedures were performed in accordance with the Declaration of Helsinki and were positively reviewed by the Ethical Committee of the University of Trento. Written informed consent was obtained. The entire session including preparation lasted ~3 h, of which 1.5 h was spent in the MEG scanner.

### Stimuli

The stimulus set consisted of videos of 14 unique ballet dancing sequences, each consisting of four smoothly connected ballet figures selected from a pool of five unique figures (i.e., passé, pirouette, arabesque, bow, and jump; see Table 2 for the structure of each unique sequence). We selected naturalistic body movements, rather than e.g., moving dots because they are rich and evolutionary salient stimuli that are likely to evoke strong neural responses and that can be modeled across a wide range from low visual to view-dependent and view-invariant levels of processing. Ballet dance was chosen to ensure

comparably smooth body movements without abrupt action boundaries. All ballet figures were presented from multiple viewpoints, to allow for studying view-dependent body posture and motion separately from view-invariant body posture and motion (i.e., orientation and position invariant). The pirouette could start and end at different angles and was always performed counterclockwise. The rationale for this stimulus design was that (1) at each point in time there should be enough stimuli similar and enough stimuli dissimilar to each other (i.e., enough variance within a model RDM at a single time point), and that (2) the model RDMs should be sufficiently dynamic (i.e., minimal temporal autocorrelation) for testing lag differences between model and neural RDMs. We recorded the dancing videos with a Canon PowerShot G9 X Mark II at a frame rate of 50 Hz (i.e., 250 frames per 5 s video). Video frames were cropped to 400 × 376 [height × width] pixels. Additionally, we recorded 3-dimensional positions of 13 passive (infrared reflecting) markers placed on the ankles, knees, hips, shoulders, elbows, wrists, and head of the dancer. Marker positions were recorded at 100 Hz using a Qualisys motion capture system (infrared-based marker capture). Markers were small silver balls, and nearly invisible in the video stimuli presented to the subjects. See https://github.com/Ingmar-de-Vries/DynamicPredictions for all 14 unique videos and temporally aligned 3D kinematic data.

### Experimental design

The experiment was created using the Psychophysics Toolbox (version 3) in MATLAB (version 2012b; MathWorks). Trials consisted of 5 s dancing videos (Fig. 4a) that were separated by blank screens with a white fixation cross presented for 1.8–2.2 s (uniform distribution). The fixation cross was also presented throughout the video, and subjects were instructed to keep fixation while covertly attending to the dancer. A single run in the MEG scanner lasted ~12 min and consisted of 90 randomly ordered sequences with the only constraint that the exact same sequence was never directly repeated (i.e., there was always at least one different sequence in between).

To ensure attention to the dance sequences, 20 trials in each run of 90 trials were two types of catch trials (Fig. 4b). In type 1 (15 trials) the sequence was interrupted at a random time between 0.5 and 5 s after onset, and subjects were asked with a yes or no question (2 response buttons) about the particular motion currently being performed (e.g., arms moving up? or pirouette?), and had to respond with a button press within 3 s. In type 2 (5 trials) the fixation cross changed from white to light purple (RGB=[117 112 179]) for 200 ms at a random time between 0.4 and 4.8 s after onset, in response to which subjects had to press a button. The color was chosen such that the change was only visible when fixating, but not from the periphery, thus stimulating

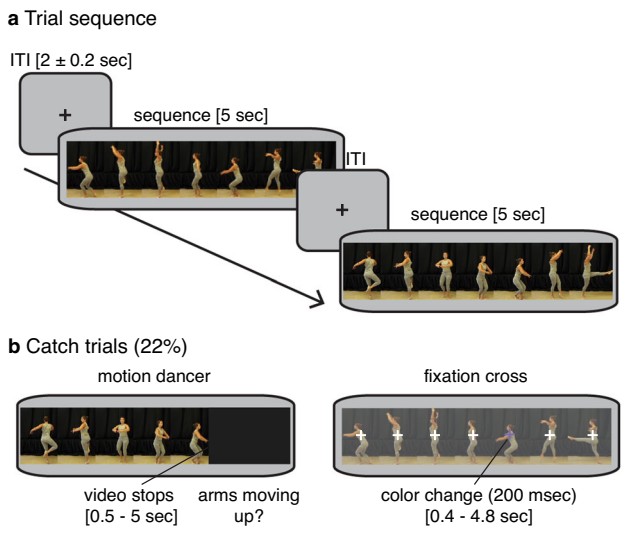

**a** Trial sequence

**b** Catch trials (22%)

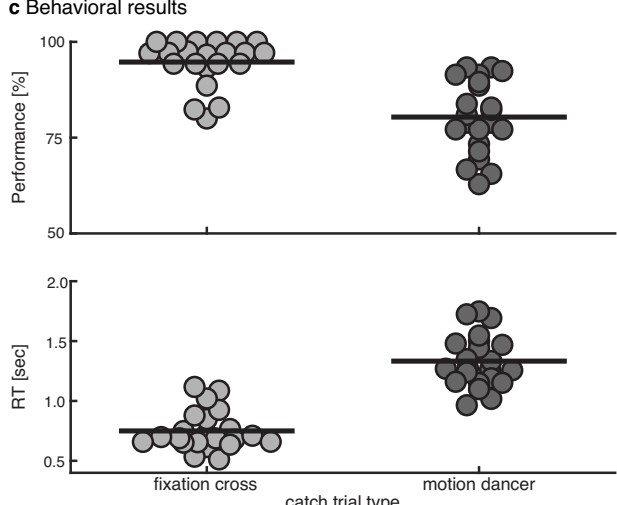

**c** Behavioral results

**Fig. 4 | Experimental design and behavioral results. a** Trial sequence. A run consisted of 90 sequences of 5 s alternated by blank screens with a fixation cross presented for 1.8–2.2 s (uniform distribution). **b** To ensure attention to the dancer, 15 trials per run were randomly interrupted between 0.5 and 5 s after onset, and subjects were asked to (dis)confirm (2 response buttons) a statement about the particular motion currently being performed (left panel). To stimulate fixation, on 5 trials per run the fixation cross changed color for 200 ms between 0.4 and 4.8 s after onset, and subjects had to press a button upon detection (right panel). **c** Behavioral results. Percentage correct and trial-averaged RT on catch trials in top and bottom panels, respectively. Dots represent single-subject data. Horizontal thick lines represent the group mean. ITI inter-trial interval.

subjects' fixation. Subjects received immediate feedback on each catch trial. In 5 out of the 15 type-1 catch trials, the question appeared only at the offset of the video (i.e., at 5 s), and these trials were therefore included in the analyses, thus leaving 75 to-be-analyzed trials per run. The remaining catch trials were excluded from all further analyses. To ensure continuous attention to the task, two subsequent catch trials were never separated by more than 8 normal trials. Subjects performed one practice run outside of the MEG scanner, in which 30% of trials were catch trials to ensure enough practice with the task. During MEG, subjects performed the behavioral task well, indicating that they attended the dancer and generally kept fixation (mean/min–max: 80/63–93% correct and 95/80–100% detection for the dancer and fixation catch trials, respectively; see Fig. 4c). Subjects performed 6.8 runs on average depending on attentional fatigue, which resulted in an average of 510 trials per subject for further analyses, with an average of 36.4 repetitions of each of the 14 unique sequences. With so many repetitions, one might expect stimulus familiarization throughout the experiment to improve prediction. However, in a post hoc dRSA analysis, we did not observe robust differences between the first and last 1/3 of the experiment (see Supplementary Discussion and Supplementary Fig. 6).

## Stimulus models

All data analyses were performed in MATLAB (version 2020a; Math-Works). We selected a total of 9 stimulus-feature models described in detail below (Fig. 5), plus a tenth subject-specific model capturing the eye position. The reason for selecting these stimulus-feature models was to capture a comprehensive characterization of the dance sequences that encompass several hierarchical levels of complexity/abstraction from low-level visual information (pixelwise) up to higher-level, perceptually more invariant information (view-invariant body posture and motion). Models based on video data (pixelwise and optical flow vectors) had a sampling frequency of 50 Hz and were therefore first interpolated to 100 Hz to match the kinematic data, using shape-preserving piecewise cubic interpolation.

Pixelwise (Fig. 5b): For each video frame, the RGB values at the 400 × 376 pixels were converted to grayscale (i.e., luminance values), smoothed (i.e., spatially low-pass filtered) using a Gaussian kernel with sigma = 5 (i.e., roughly 11.75 pixels full width at half max[18]), as implemented in MATLAB's `imgaussfilt.m` function, and vectorized (i.e., 150,400 feature vector per frame). For each frame separately, these vectors were then correlated between two different videos, to generate a single entry in the model RDM, where dissimilarity was defined as 1-Pearson correlation. This operation was repeated for each of the possible stimulus pairs to generate a 14 × 14 RDM at a single frame and subsequently repeated over the 500 frames to generate 500 model RDMs (see Fig. 1e for an example).

Optical flow vectors (Fig. 5c): We converted videos to grayscale and used MATLAB's `estimateFlow.m` implementation of the Farne-bäck algorithm[41] to compute optical flow vectors, which capture the apparent motion between subsequent frames at each pixel. These optical flow vectors were spatially smoothed using a Gaussian kernel[42], with the same settings as for the pixelwise grayscale model, and subsequently divided into magnitude and 2D direction, and vectorized (i.e., 150,400 and 300,800 feature vectors for magnitude and direction, respectively). Model RDMs were computed as described for the pixelwise model.

View-dependent body posture (Fig. 5d): We defined view-dependent body posture as the 3D positions of the 13 kinematic markers (i.e., 39 feature vector), relative to a coordinate system with origin placed on the floor in the center of the video frame. Model RDMs were computed as described for the pixelwise model.

View-invariant body posture (Fig. 5d): For the view-invariant body posture model, before computing the dissimilarity between two postures in two different stimuli, we used a modified version of MATLAB's `procrustes.m` function to align the 3D marker structure of one stimulus with the marker structure of the other stimulus, while keeping its internal structure intact, by using translation, and rotation along the vertical axis only, to minimize the sum of squared differences between the two 3D marker structures. For instance, if in two to-be-compared videos the exact same ballet figure is performed but in opposite direction, this would result in high dissimilarity in terms of view-dependent body posture, but high similarity in terms of view-invariant body posture. Since the direction of alignment matters for dissimilarity (i.e., dissimilarity(frame1,frame2aligned) ≠ dissimilarity(frame1aligned,frame2)), we computed dissimilarities after alignment in both directions, and subsequently averaged over them.

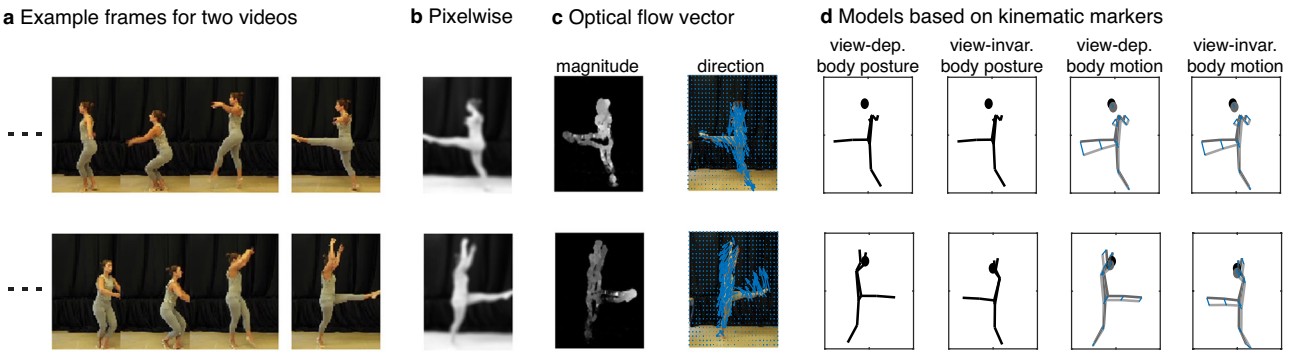

**a** Example frames for two videos  **b** Pixelwise  **c** Optical flow vector  **d** Models based on kinematic markers

**Fig. 5 | Illustration of stimulus models. a** Example frames of two videos (rows) to be correlated for creating the model representational dissimilarity matrices (RDMs). From left to right; **b** the pixelwise smoothed grayscale of the video frame, **c** the magnitude of the optical flow vectors (brighter color is higher magnitude) and the direction of the optical flow vectors (as indicated by blue arrows, which are scaled 10 times here for illustrative purposes), and **d** view-dependent body posture (as defined by 3D kinematic marker positions), view-invariant body posture (i.e., after aligning the kinematic markers of the second video with those of the first video without changing the internal structure, through translation and rotation along the vertical axis), view-dependent body motion as indicated by blue lines (i.e., difference in kinematic marker position between two subsequent frames), and view-invariant body motion.

View-dependent body motion (Fig. 5d): View-dependent body motion was operationalized as the first derivative of view-dependent body posture, i.e., the difference in the 3D marker positions between two subsequent frames: $v_m(i) = x_m(i) - x_m(i - 1)$, where $v$ is the motion of marker $m$ in frame $i$, and $x$ is the marker position. Body motion vectors, therefore, had a magnitude and a 3D direction. Motion time series were shifted and interpolated such that the motion calculated in the formula above for frame $i$ was placed in between frame $i$ and frame $i-1$. Model RDMs were computed as described for the pixelwise model.

View-invariant body motion (Fig. 5d): View-invariant body motion was computed the same way as view-dependent body motion, but after aligning one body posture to the other as described above for view-invariant body posture. Model RDMs were computed as described for the pixelwise model.

Eye position: Since eye movements are a common problematic covariate in neuroimaging studies using multivariate approaches[43], even if like here a fixation task is used (e.g., micro-saccades remain), we added individual-subject RDMs based on eye-position as a control to ensure that the dRSA results for our models of interest were not due to eye movements. The raw eye tracker signals were first downsampled to 100 Hz, after which trials in which the eye tracker lost the eye for more than 10% of the samples were removed (a total of 1.0% of trials was missing; min–max across subjects = 0–5.9%), and the remaining missing samples were interpolated (e.g., due to blinks; a total of 0.3% of samples; min–max across subjects = 0–2.1%). Finally, signals were averaged over all remaining repetitions of the same unique stimulus. Eye position models were subject-specific. Model RDMs were computed as described for the pixelwise model, except that dissimilarity between eye positions was defined as the Euclidean distance between the four eye position values (i.e., $x$-, and $y$-position for the left and right eye), as four values are too low for a reliable correlation estimate. Note that while there was a representation of eye position present in the neural signal (Fig. 2), this was successfully regressed out when testing each of the other models (as confirmed by the simulations; see subsection Simulations). Therefore, eye movements could not explain any of the findings for the other models.

View-dependent and view-invariant body acceleration were operationalized as the second derivative of marker position (i.e., the first derivative of marker motion). These models only explained a minimal amount of variance in the neural data and are therefore not illustrated in the main results. However, because these models did correlate with some of the tested models (see subsection Simulations), we regressed them when testing each of the other models (see subsection Dynamic representational similarity analysis for details on regression).

## MEG data collection and preprocessing

We obtained whole-head MEG recordings at a sampling rate of 1000 Hz using a 306-channel (204 first-order planar gradiometers, 102 magnetometers) VectorView MEG system (Neuromag, Elekta) in a two-layer magnetically shielded room (AK3B, Vacuum Schmelze). A low-pass antialiasing filter at 330 Hz and a DC offset correction was applied online. Importantly, no online or offline high-pass filter was applied, since high-pass filtering can temporally spread multivariate information[44]. Before the MEG recording, we digitized the individual head shape with an electromagnetic position and orientation monitoring system (FASTRAK, Polhemus) using the positions of three anatomic landmarks (nasion and left and right preauricular points), five head position indicator coils and 300+ additional points evenly spread on the subject's head. Landmarks and head-position induction coils were digitized twice to ensure a localization error of <1 mm. To co-register the head position in the MEG helmet with anatomic scans for source reconstruction, we acquired the head positions at the start of each run by passing small currents through the coils. Binocular eye movements were recorded at 1000 Hz using an SR-Research Eyelink Plus eye tracker. In addition, horizontal and vertical EOGs were recorded from 2 electrodes located 1 cm lateral to the external canthi and 2 electrodes located 2 cm above and below the right eye, respectively. Stimuli were presented in full color using a VPixx PROPixx projector. All hardware was connected to a DataPixx input/output hub to deliver visual stimuli and collect MEG data, eye-tracker data, and button presses in a critical real-time manner (VPixx Technologies), and stimulus-onset triggers were directly stored together with the MEG data. In addition to these triggers, we also placed a photodiode in the top right of the screen (i.e., outside of the video stimulus frame), and changed a square on the screen underneath the photodiode from black to white at the onset of each single video frame, as to realign MEG signals to these photodiode signals offline.

MEG data were preprocessed offline using a combination of the Brainstorm (version 3)[45], Fieldtrip (version 20191113)[46] and CoS-MoMVPA (version 1.1.0)[47] toolboxes in MATLAB (version 2020a; MathWorks), as well as custom-written MATLAB scripts (shared at https://github.com/Ingmar-de-Vries/DynamicPredictions). Continuous data from each run were visually inspected for noisy sensors and system-related artifacts (e.g., SQUID jumps), and a maximum of 12 noisy sensors were removed and interpolated. Next, we applied the Neuromag MaxFilter (version 2.2) implementation of Signal Source Separation (SSS)[48] for removing external noise from each individual run and spatially aligned the head position inside the helmet across runs. After these initial steps, data were loaded into Brainstorm, where consecutively spatial co-registration between anatomical MRI and

head position in the MEG helmet was refined using the 300+ additionally registered points on the head (see subsection Source reconstruction), data were filtered for line noise using a band-stop filter at 50 and 100 Hz (0.5 Hz bandwidth), data were down-sampled to 500 Hz, and independent component analysis (ICA) was applied separately for magneto- and gradiometers in order to detect and remove an average of 1.2 blink and eye movement components and 0.6 cardiac components. Blink and eye movement components were confirmed by comparing their time series with the EOG channels. ICA was applied to a temporary version of the data that was first down-sampled (250 Hz) and band-pass filtered (0.1–100 Hz) to reduce computation time and improve ICA, respectively. The ICA weight matrix was subsequently applied to the original data. Note that the maximum number of ICA components to be extracted was determined by the residual degrees of freedom after SSS rank reduction during MaxFiltering. Continuous data were then segmented in epochs from −1.5 to 6.5 s locked to the onset of the video stimulus, according to the triggers stored with the MEG data, and single epochs were DC offset corrected using a baseline window of −500 to 0 msec relative to stimulus onset. Each epoch was visually inspected, and those containing artifacts (e.g., muscle activity, SQUID jumps, etc.) were discarded from further analyses, which resulted in a rejection of 7.2% of all trials. After these steps, data were exported from Brainstorm into the Fieldtrip data format, epochs were realigned to the photodiode onset for higher temporal precision, data were temporally smoothed using a 20 ms boxcar kernel as implemented in Fieldtrip's `ft_preproc_smooth.m`, downsampled to 100 Hz, and finally averaged over all ~36 repetitions of the same video stimulus. Note that we opted for transforming MEG signals to source space (see next section), but if one would opt to stay in sensor space instead, we recommend applying multivariate noise normalization (MNN[49]).

## Source reconstruction and ROI selection

For the construction of 3D forward models, we first acquired previously recorded anatomical 3D images if available (13 subjects) and used the ICBM152 standard brain for the remaining 9 subjects. Individual anatomical images were obtained using a Siemens Prisma 3 T, actively shielded, whole body 3 T magnet. A few older individual anatomical scans already present at the facility were collected with the former scanner, i.e., 4 T MRI scanner (Bruker Biospin). Fiducial points were marked automatically (MNI) and inspected, after which the anatomical scans were 3D reconstructed using CAT12 (version 2170), a computational anatomy toolbox[50]. For the 9 subjects without individual anatomy, the standard anatomy was warped to the subject's head volume as estimated from the +300 digitized head points. Next, head models were computed by coregistering the subjects head shape with the reconstructed MRI brain volumes using Brainstorm's default settings and using overlapping spheres, after which we performed source reconstruction using minimum-norm estimates (MNE) for a source space of 15,000 vertices[51], as implemented in Brainstorm. We used both gradio- and magnetometers. For MNE, the first noise covariance matrices were computed after single-trial DC offset correction and using the time window from −1000 to 0 ms relative to stimulus onset. Second, the data covariance matrices were computed using the −1000 to 0 ms interval for baseline correction, and the 0 to 5000 ms interval (i.e., stimulus presentation) as data interval. Last, we computed non-normalized current density maps as a measure of source activity, with source direction constrained to be normal to the cortex (i.e., a single source activity value at each vertex).

The 15,000 vertices were parcellated into 360 parcels according to the cortical atlas from the Human Connectome Project (HCP[52]), which provides the most precise parcellation based on the structural and functional organization of the human cortex to date. We a priori selected six regions of interest (ROIs; Fig. 6) that consisted either of a single parcel from the atlas (in case of V1 and V2), or a combination of several parcels (in case of V3 + V4, the lateral occipitotemporal cortex (LOTC), the anterior inferior parietal lobe (aIPL), and the ventral premotor cortex (PMv)). Besides the purely visual areas, we selected the LOTC, aIPL and PMv because they form the action observation network (AON[53]) and were expected to be important for processing and predicting these specific stimuli[31,54]. This selection resulted in comparably sized ROIs with on average 484 vertices (min–max = 403–565), of which the vertex signals were used as features for computing the neural RDM for a given ROI. The neural RDM was computed as implemented in cosmoMVPA[47], on the centered data, using 1-Pearson correlation as a dissimilarity measure.

## Dynamic representational similarity analysis

As a first step, neural and model representational dissimilarity matrices (RDMs) were computed for each time point as described above (see also Fig. 1), resulting in neural and model RDMs of 91 features (14 × 14 stimuli gives 91 unique values) by 500 time points. To improve the SNR, neural and model RDMs were temporally smoothed using a 30 ms boxcar kernel as implemented in Fieldtrip's `ft_preproc_smooth.m`. To compute the dRSA, the similarity between neural and model RDMs was estimated at each neural-by-model time point, thus resulting in a 2-dimensional dRSA matrix (Fig. 1f; lower panel). This 2D dRSA matrix was further averaged over time for each neural-to-model time lag (i.e., time-locked to the on-diagonal) within the range of −1 to 1 s, thus resulting in a lag-plot (Fig. 1; upper panel), in which a peak on the right of the vertical zero-lag line indicates that the neural RDM is most similar to a model RDM earlier in time (i.e., it is lagging behind the stimulus information), whereas a peak on the left of the vertical zero-lag line indicates that the neural RDM is predicting a future model RDM. One problem with using a simple correlation as a similarity measure for RSA is that different stimulus models might share variance (e.g., view-dependent and invariant body posture will inevitably share some variance), making it impossible to unambiguously assign an RSA result to one specific model. Additionally, specific to dynamic RSA, model RDM $X$ at $t_1$ might share variance with model RDM $Y$ at $t_2$ (e.g., body motion will correlate with body posture at an earlier and later time point), and model RDM $X$ might additionally share variance with itself at distant time points due to temporal autocorrelation. These problems are illustrated below for simulated data (see subsection "Simulations" in the "Methods" section). For completeness, the dRSA results using simple correlations are illustrated in Supplementary Fig. 2. While these results (Supplementary Fig. 2) do reflect truly shared variance between neural and model RDMs, caution is warranted with their interpretation as it is difficult to separate the contribution of different model RDMs. Note that both shared variance between models and temporal autocorrelation within models are not a problem with the dRSA approach per se, but rather inherent to naturalistic (dynamic) stimuli. We argue that our followed approach explained in the next paragraph offers a solution to separate the contribution of different model RDMs.

In order to extract variance from the neural data uniquely explained by a specific stimulus model, while disregarding variance better explained by other models, we used linear least-squares regression to estimate similarity. Here, the neural RDM at a certain time point $t_y$ was the response variable $Y$ (size 91 × 1), our model RDM of interest at a certain time point $t_x$ was one predictor variable $X_{test}$ (size 91 × 1), and all to-be-regressed out RDMs were other predictor variables $X_{regressout}$ (size 91 x N), with our measure of similarity being the regression weight for $X_{test}$. As to-be-regressed out RDMs we selected the other 9 model RDMs at each time point (i.e., size 91 × $N_{time}$ per model), plus the model RDM of interest itself at distant time points (i.e., at a minimum distance from $t_x$ at which the model shared less than 10% variance with itself at $t_x$; determined by simulations; see subsection "Simulations" in the "Methods" section). This would result in >1800 predictor variables in $X_{regressout}$ (i.e., size 91 × 1800). However,

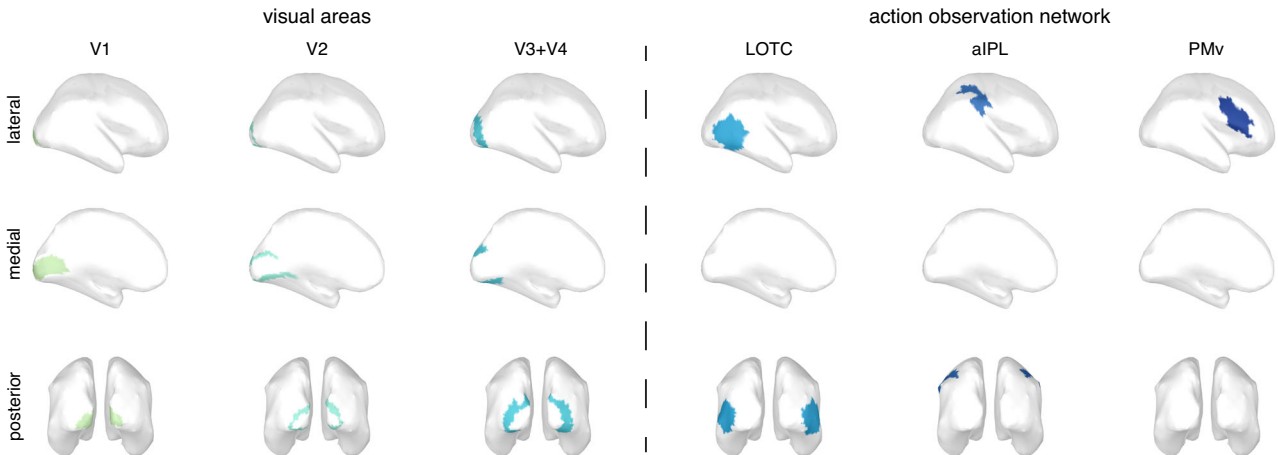

**Fig. 6 | ROI definitions.** Cortical regions of interest for main ROI analysis (Fig. 2), based on combinations of parcels as defined according to the Human Connectome Project (HCP) atlas[52]. Here ROIs were a priori defined as follows (ROI name = atlas parcels): V1 = V1 [487 vertices]; V2 = V2 [491 vertices]; V3 + V4 = V3 and V4 [490 vertices]; LOTC = V4t, FST, MT, MST, LO1, LO2, LO3, PH, PHT, TPOJ2 and TPOJ3 [565 vertices]; aIPL = PF, PFt, AIP and IP2 [403 vertices]; PMv = IFJa, IFJp, 6r, 6 v, PEF, IFSp, 44 and 45 [466 vertices]. Note that these vertex amounts are based on the ICBM152

template cortical surface, exact amounts differ slightly between individual subjects. MEG responses at all vertices from both hemispheres were combined into a single vector to compute the neural RDM at a single time point (i.e., pairwise dissimilarity in the MEG response to the 14 sequences). For visualization, all vertices within a single ROI are given the same color, which matches the color for each ROI in Fig. 2. LOTC lateral occipitotemporal cortex, aIPL anterior inferior parietal lobe, PMv ventral premotor cortex.

the amount of predictor variables in linear least-squares regression is limited by the number of features (also called observations; here a maximum theoretical value of 91, but one should stay below this maximum to prevent overfitting). We, therefore, used principal component analysis (PCA) across two steps to reduce the dimensionality of $X_{\text{regressout}}$. First, we ran PCA separately per to-be-regressed out model (size $91 \times N_{\text{time}}$) and selected only those principal components capturing at least 0.1% of total variance from the $N_{\text{time}}$ input variables. Next, the new $X_{\text{regressout}}$, now consisting of principal components, was combined with $X_{\text{test}}$, after which a second PCA was run over this combination. From this second PCA, only the first 75 components (ordered by total explained variance) were selected (limited by the maximum of 91 predictors), and these components were used as predictor variables in the linear least-squares regression. Subsequently, the principal component regression weights were projected back onto the original predictors using the PCA loadings, in order to specifically extract the regression weight for $X_{\text{test}}$. This approach, termed principal component regression (PCR), has the advantage that it strongly reduces the number of predictors while maintaining most of their total variance, and the advantage that the PCA components entered into the regression are decorrelated, hence preventing multicollinearity. Before each PCA, individual model RDMs and the neural RDM were rescaled (to min-max: 0–2) at once for all time points, to equalize scales between RDMs, while keeping the temporal changes in scale intact that are inherent to a certain model. A large difference in scale between models could cause early PCA components to simply pick up the model with the largest scale, which arbitrarily depends on dissimilarity measure (i.e., Euclidean distance used for the eye-tracker RDM has a much larger scale than corr-1 used for the other model RDMs). Additionally, the RDM was centered per individual time point before each PCA and regression. Importantly, PCR was effective at extracting a model representation of interest while regressing out other model representations with which it might share variance, as confirmed by simulations (see subsection Simulations).

**Temporal subsampling**
Our dRSA pipeline involved one additional important step to increase the generalizability of the results as well as the SNR. Imagine that at a given time point $t_x$, in all 14 videos, the ballet dancer is in the middle of a figure and moving relatively slowly and predictably. At $t_x$ prediction is

possible far in advance, resulting in an early predictive dRSA peak. In contrast, if at $t_y$ there is a transition between ballet figures in all 14 videos, prediction is not possible very far in advance, resulting in a later (or no) predictive dRSA peak. Last, if at $t_z$ stimulus predictability is different for each of the 14 videos (likely the most occurring scenario), the predictive peak reflects an average thereof. In other words, stimulus-specific feature trajectories influence how far in advance a stimulus can be predicted, and thus the structure of the neural and model RDMs depends on the exact arbitrary pairwise alignment of the 14 ballet dancing sequences (and the alignment of event boundaries at different timescales[55], such as onsets and offsets of individual ballet figures, subfigures, smaller motions, etc.). Consequently, the pattern across stimulus time in the 2-dimensional dRSA matrix is idiosyncratic and heterogeneous. If videos would have started a little earlier or later into the first ballet figure, the structure of RDMs, and therefore of the dRSA matrix, would look completely different at all time points. We are not interested in these specific 14 stimuli and their pairwise temporal alignment per se, but rather in dynamic prediction more generally. Ideally one would have an infinite number of stimuli, i.e., infinitely large RDMs.

To mimic an infinite number of stimuli, and thus minimize the idiosyncrasy of our dRSA results and increase SNR, we applied a temporal subsampling and realignment approach. Specifically, across 1000 iterations, a 3-s segment was randomly extracted from each of the 14 5-s stimuli (Fig. 7a; orange boxes). Next, these 14 new 3-s segments were realigned to each other, and RDMs were computed at each realigned time point $t_R$ (Fig. 7b, c). Last, the similarity between neural and model RDMs was computed for each realigned neural by model time, resulting in the 2-dimensional dRSA matrix. Crucially, within a single iteration, while a different random 3-s window was selected for each of the 14 stimuli, for a given stimulus the temporal alignment between neural and model data remained intact (Fig. 7a; dotted vertical orange lines). This subsampling approach accomplishes a different pairwise alignment of the 14 stimuli on each iteration, thus creating completely new RDMs and temporal structures in the dRSA results. This is illustrated in Fig. 7d: First, the dashed black line illustrates the results for V1 without the temporal subsampling step (taking the whole 5-s videos; see Supplementary Fig. 3 for the complete results). Second, the light gray lines illustrate the dRSA results on an example of 10 iterations. There is quite some variability in the shape of the curves, as

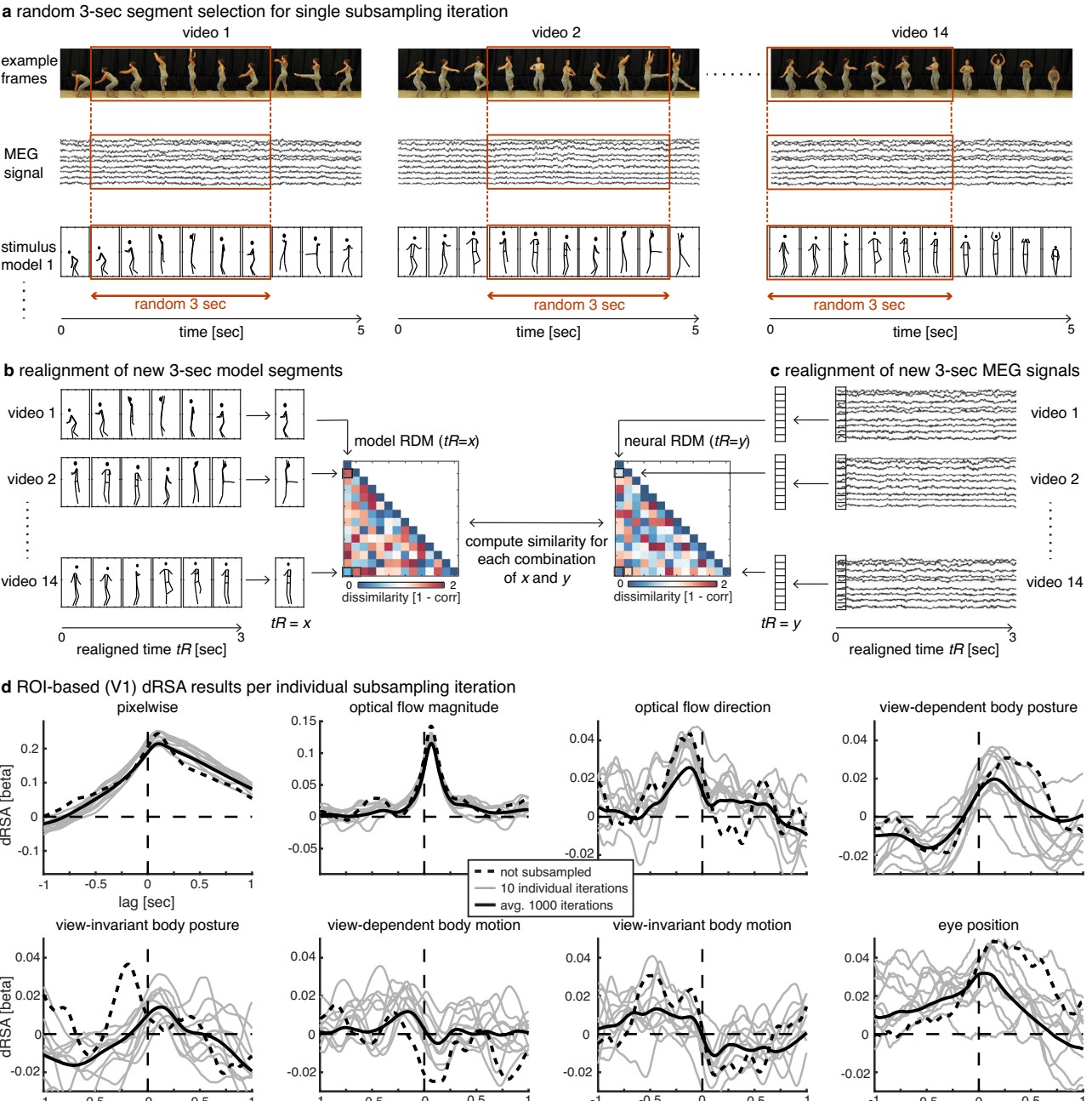

**Fig. 7 | Temporal subsampling and realignment. a** On each subsampling iteration, a 3-s segment is randomly selected independently for each of the 14 stimuli (orange boxes). For a given stimulus, the same 3-sec window is selected for both neural and model data, thus keeping temporal alignment between those intact (indicated by vertical orange dotted lines). **b** and **c** Next, the 14 new 3-s segments are realigned, after which model and neural representational dissimilarity matrices (RDMs) are computed at each realigned time point $t_R$. Last, similarity is computed for each combination of realigned model time ($x$) and neural time ($y$). Note that the steps in **a**–**c**, as well as the last step to compute the dynamic representational similarity analysis (dRSA) plot, are all done within a single subsampling iteration. After 1000 subsampling iterations, dRSA results are averaged over iterations. **d** Region of interest (ROI)-based dRSA results for V1. The dashed black line indicates dRSA results for the exact same analysis as presented in Fig. 2a, but without the temporal subsampling and realignment steps, i.e., dRSA analysis was performed directly on the original 5-s stimuli (see Supplementary Fig. 3 for other ROIs and statistical significance). The gray lines indicate single 3-s subsampling iterations, while the full black line indicates the average over 1000 subsampling iterations (equal to V1 in Fig. 2a).

well as the peak timings, which are dependent on the arbitrary pairwise stimulus alignment on that given iteration. Last, the full black lines indicate the dRSA results after averaging the 1000 iterations. Note that this whole procedure is done on an individual subject level, and the averaged dRSA values are subsequently used for statistical analysis (see subsection Statistical analysis). To summarize, this approach mimics having many more video stimuli that are differently aligned to each other, increasing SNR, and making the results become more generalizable (at least to other sets of ballet dancing videos).

Note that to successfully apply dRSA, it is worth considering a few criteria that the model RDMs ideally fulfil. Specifically, they need to be sufficiently large (i.e., have enough unique stimuli) to allow for enough regressors to be included in the PCR, sufficiently large such that results are generalizable rather than specific to the stimulus set (which we solved here with the temporal subsampling and realignment procedure), and dynamic enough that the structure of the RDM at different time points is distinguishable (i.e., stimulus pairs need to be similar at some times while dissimilar at other times).

## Statistical analysis

Before group-level statistics (and plotting), we Fisher Z-transformed all beta weights to get normally distributed values[56]. Because statistical testing of the dRSA lag-plots involves many comparisons (at each lag time-point), we performed group-level nonparametric permutation testing with cluster-based correction for multiple comparisons[57], as implemented in Fieldtrip's `ft_timelockstatistics.m` function, with the method parameter set to "montecarlo". We used t-values from a one-sided t-test as test statistic (i.e., to test whether beta weights are significantly larger than zero), we used the sum over the t-values belonging to a cluster of significant time points as a measure of cluster size, and we ran the permutation 25,000 times for each analysis. We selected a lenient and strict threshold for significance, with a single-sample threshold at $p < 0.01$ or $p < 0.001$, combined with a cluster-size threshold at $p < 0.05$ or $p < 0.01$, respectively. In the main results (Fig. 2a), lenient significant intervals are indicated by thick horizontal bars with colors matching the respective ROI line plots, while strict significant intervals are indicated with thin black or white horizontal bars inside the colored bars. Only significant clusters larger than what can be expected by chance survive this procedure.

## Simulations

The goal of the simulations was to find a dRSA pipeline that was effective at testing only a single model of interest while regressing out other models with which the model of interest might share variance, and while regressing out the model of interest itself at distant (future or past) time points (i.e., to reduce effects of temporal autocorrelation). Principal component regression (see subsection Dynamic representational similarity analysis) appeared most effective at achieving this goal. Note that to prevent double-dipping[58], we first fine-tuned the dRSA pipeline described above on the simulated data, before subsequently applying the exact same pipeline to the real neural data.

Each simulated neural RDM consisted of a single model RDM at zero lag. We first ran dRSA using a simple correlation for each tested model, i.e., the correlation coefficient was used as a similarity measure (Fig. 8a, see also Supplementary Fig. 5 for shared variance, i.e., correlation squared). Note the nonzero off-midline correlations, which confirm a complex relationship between and within models across time. This complex structure of shared variance limits unambiguous inference if the simple correlation is used for dRSA. Subsequently, the exact dRSA pipeline as described above using PCR was applied on each of the simulated neural RDMs (Fig. 8b), which indicated that PCR is effective in extracting only the model of interest from the simulated neural RDM, while largely regressing out the other models. Note that not all models are completely regressed out, which is most obvious when motion models are tested on simulated data in which acceleration models are implanted. However, it is important to note that shared variance with another model is only problematic if that other model itself explains variance in the neural data. In other words, if model $X$ explains variance in the neural data, and model $Y$ does not, the observed results for model $X$ cannot be explained by it sharing variance with model $Y$. Importantly, any of the remaining variance of a to-be-regressed out model $Y$ that we observed while testing another model $X$ (e.g., acceleration when testing motion; Fig. 8b), could not explain the main results for model $X$ (e.g., motion; Fig. 2), as model $Y$ itself did not explain enough of the neural data (this is the case for all models that have not been fully regressed out).

We performed two additional simulations to further test the effectiveness of PCR. First, we combined the first 3 models (pixel-wise, optical flow magnitude, and optical flow direction) with a random dynamic neural RDM (weight ratio 1:1:1:3) to create a single simulated neural RDM (Fig. 8c). The random neural RDM consisted of $91 \times N_{time}$ random values drawn from a uniform distribution, such that there was no temporal autocorrelation. As in the main dRSA pipeline, we tested

each of the models separately, while regressing out the other 9 using PCR. The rationale for selecting the first 3 models was that the first two models (pixel-wise and optical flow magnitude) have a stronger representation in the real neural signal (Fig. 2), whereas the third model (optical flow direction) shows a predictive representation. Additionally, as all three are based on video data, they might be more likely to share variance. We wanted to ensure that the predictive representation in the real neural signal for optical flow direction could not be caused by the presence of a representation for the first two models in the neural signal. Importantly, this analysis confirmed that PCR is effective in extracting a single model from the combined simulation while regressing the other two. Second, we ran dRSA on optical flow direction as a simulated and tested model, but with different levels of added random noise (1:0, 1:1, 1:5, and 1:10) to create the simulated neural RDM (Fig. 8d). This analysis showed that simply adding more noise (i.e., variance unrelated to the model RDM of interest), can explain the values typically observed in the real neural data (Fig. 2).

Additionally, using the simulated data we explored several, perhaps more tractable, alternative approaches such as partial correlation to deal with the large amount of (covarying) predictor RDMs. Two example alternatives are illustrated in Supplementary Fig. 4. For the first approach based on partial correlation, peak latencies of the cross-model correlations were extracted from Fig. 8a, after which a partial correlation that included the model of interest at a certain timepoint $t_x$, as well as all other models at their respective peak latencies in the cross-correlation with the model of interest (Supplementary Fig. 4b). Since some cross-model correlations displayed two peaks (e.g., view-invariant body posture and motion; see Fig. 8a), a maximum of 2 peak latencies were selected for each to-partial-out model. The second approach based on partial correlation included besides the model of interest all other models after downsampling (250 ms; Supplementary Fig. 4c). The simulations illustrated in Supplementary Fig. 4 indicate that both alternatives perform worse at attenuating effects of co-varying models, and in some cases, they produce quite extreme (spurious) negative values, likely due to multicollinearity.

## Peak latency

Peak latencies were defined as the timing of the highest peak in the dRSA curves (Fig. 2a), separately for the negative (−1 to 0 s) and positive (0 to −1 s) intervals for the optical flow direction model (as we observed two distinct peaks), or at once for the whole −1 to 1 s time interval for all other models. Given that peak latencies of the dRSA curves are difficult to estimate on a single-subject level (i.e., due to noise causing multiple peaks, similar to ERP/ERF analyses), we report the peak latencies of the subject-average dRSA curves in Table 1 and used retrieved jackknifed peak latencies for statistical analyses[23,24]. Specifically, across 22 iterations we excluded one subject from the dataset, averaged dRSA curves over the remaining 21 subjects, and estimated the peak latency for this average dRSA curve, until each subject was excluded once, thus resulting in 22 jackknifed peak latencies. A problem with jackknifing is that because each jackknifed value now represents the average over $n−1$ subjects, the standard deviation across these values is artificially small. In order to use these jackknifed values in standard parametrical tests, we retrieved single-subject latency estimates by subtracting each individual jackknifed value multiplied by $n−1$, from the grand average jackknife value multiplied by $n$[24]. Essentially, rather than representing true individual subject dRSA peak latencies, these individual jackknifed values are a measure of how much a given subject's dRSA curve influenced the subject-average peak latency, while their average is a good estimate of the subject-average peak latency (Table 1). Based on visual inspection of the main dRSA curves (Fig. 2a), we performed two sets of posthoc analyses on these jackknifed peak latencies. First, to test for a difference between pixelwise and optical flow magnitude, we performed an

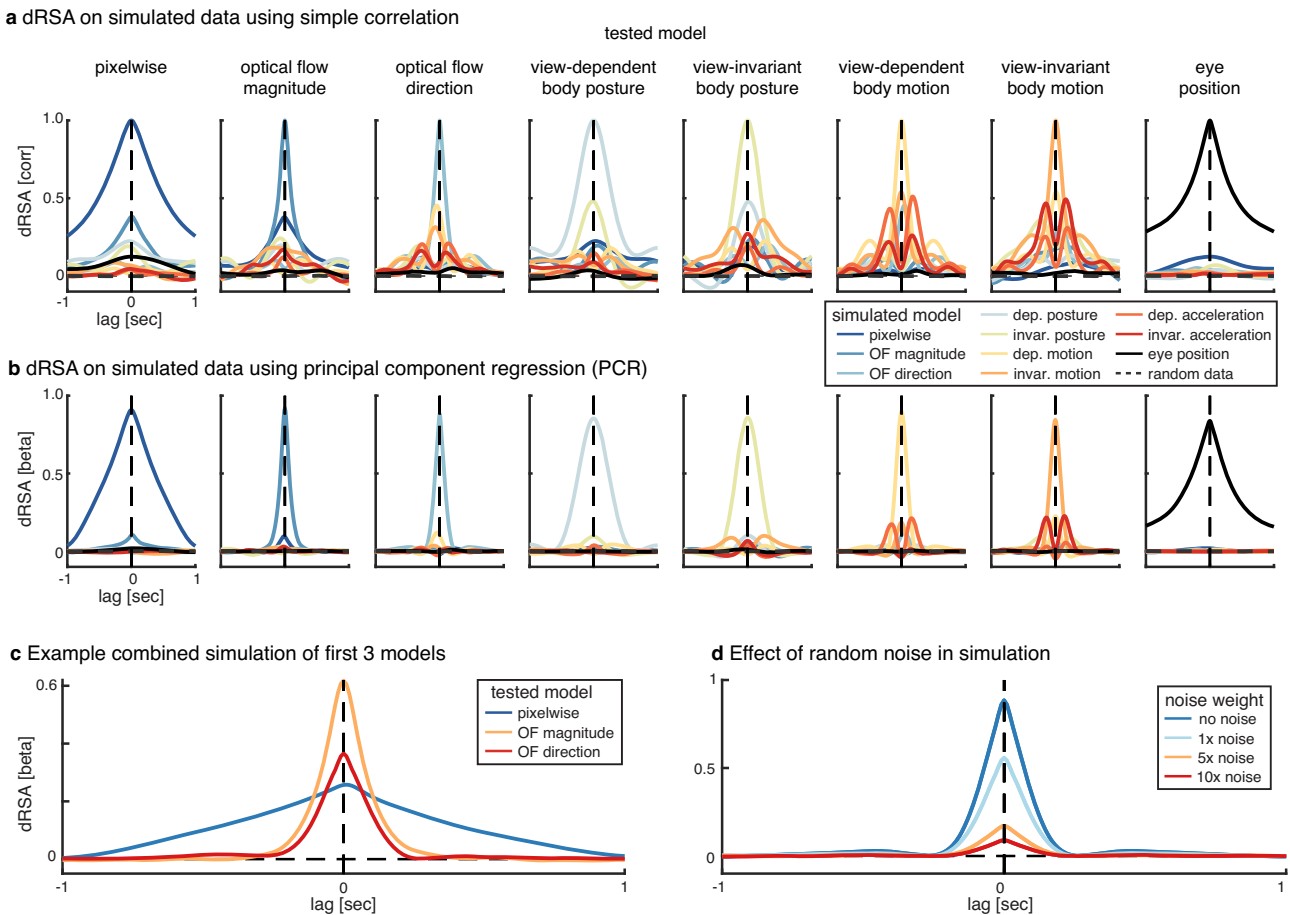

**Fig. 8 | Simulations. a** Correlation as similarity measure for dynamic representational similarity analysis (dRSA). Colors indicate separate neural simulations of individual model representational dissimilarity matrices (RDMs). Columns indicate which model was tested. These results illustrate the effect of shared variance between various models and the effect of temporal autocorrelation within a given model. **b** Regression weight as similarity measure for dRSA using principal component regression (PCR; see the "Methods" section for details). In short, simulated neural RDMs are exactly the same as in (**a**), but at the final step in dRSA (i.e., comparing neural and model RDMs) all models are included in a single regression, and only the beta weight of the tested model (columns) is illustrated. Importantly, these results indicate that PCR is effective at extracting only the model of interest from the simulated neural RDM, while largely regressing the other models. **c** The first 3 models (pixel-wise, optical flow magnitude, and direction) are combined to create a single simulated neural RDM. In each dRSA analysis, only a single model is tested (colors), while the others are regressed out using PCR (as in **b**). Importantly, PCR is effective in extracting a single model while regressing the other two. **d** Same PCR analysis as (**b**) and (**c**), with optical flow direction as simulated and tested model, but with different implant-to-random noise ratios to create the simulated neural RDM.

ANOVA with factors model (2) and ROI (4), where we included only the first 4 ROIs for which dRSA values were significant in the main analysis (Fig. 2a), as one could argue that peak latencies are unreliable if the dRSA curve itself is not significant. Second, we performed a similar ANOVA with factors model (3) and ROI (4), to test for a difference in predictive peak latency for optical flow direction, view-dependent body motion, and view-invariant body motion. Effect sizes were computed as partial eta squared ($\eta_p^2$).

**Representational spread (RS)**

The pyramidal shape of dRSA curves (Fig. 2a) could suggest a temporally extended (i.e., sustained) neural representation that is not completely overwritten by new visual input from subsequent video frames. However, this shape is partly explained by the temporal autocorrelation of models (Fig. 8a), which also affects the shape of the dRSA curve resulting from the PCR method (Fig. 8b). We quantified the relative amount of information spread surrounding a dRSA peak, above and beyond what can be accounted for by the model autocorrelation, by means of a representational spread index (hereafter RS; Fig. 3a). Specifically, we first normalized for each model both the observed and the simulated dRSA curves by dividing each by its maximum (i.e., peak) value. Next, we aligned the curves such that they

both had a peak latency of zero and a peak value of 1. Last, we subtracted the simulated curve from the observed curve to retrieve the RS. As the RS relies on an accurate estimate of dRSA peak, we followed the exact same jackknife procedure described above for peak latencies. For this analysis, the simulated dRSA curve was computed using the PCR pipeline as for Fig. 8b, thus capturing the pure effect of the model autocorrelation in terms of temporal information spread in the dRSA curve. Note that there is another source of autocorrelation, i.e., in the neural data. However, one needs to remember that the neural RDM is computed by first averaging ~36 trials of the exact same video, before computing pairwise dissimilarity between all 14 stimuli. By averaging over ~36 trials any neural autocorrelation that is not stimulus-related is canceled out. Any remaining neural autocorrelation is thus related to the stimuli, which is exactly what we are interested in. That is, if this stimulus-related neural autocorrelation is stronger than can be explained by the model autocorrelation, then it must be that the neural signal contains a spread-out representation of the model.

**Searchlight on cortical surface**

To explore the spatial dRSA pattern across the cortical surface, we performed a whole-brain analysis. For this analysis, we parcellated the 15,000 sources in 400 parcels according to the Schaefer atlas[59], for

which the parcels are more similar in the number of vertices compared to the HCP atlas (i.e., average ± std for HCP and Schaefer400 are 40 ± 33 and 36 ± 17, respectively). For each parcel, we computed dRSA as described above. To reduce computation time, we only computed dRSA at the neural-to-model time lag at which we observed a peak in the main ROI-based results (Fig. 2a), resulting in 9 separate searchlight analyses (i.e., 2 peaks for optical flow direction, and 1 peak for all other models; see Fig. 2b). For creating the cortical source maps, all vertices belonging to a certain parcel were given the same dRSA beta weight, hence giving each of 15,000 sources a dRSA weight. These values were subsequently loaded into the Brainstorm GUI database and spatially smoothed using a Gaussian kernel with a full width at half maximum of 2 mm. We tested whether the dRSA beta weight at each of the 400 parcels was significantly larger than zero using one-sided $t$-tests and used FDR correction at $p < 0.05$ to correct for multiple comparisons across the 400 parcels. Figure 2b illustrates the FDR-corrected source maps.

### Reporting summary
Further information on research design is available in the Nature Portfolio Reporting Summary linked to this article.

### Data availability
The raw and processed MEG data are available under restricted access, as the authors are still using this MEG dataset for a successive project. Access can be obtained by request to the corresponding author. As soon as the authors are finished with the dataset, it will be made publicly available. However, note that a complete dataset of a single subject is already available at OSF such that the analysis pipeline can be tested (https://doi.org/10.17605/OSF.IO/ZK42F). Source data are provided with this paper, and available on the OSF at https://doi.org/10.17605/OSF.IO/ZK42F. Last, this study made use of two open-source atlases implemented in Brainstorm: The Human Connectome Project (HCP) atlas[52] and the Schaefer atlas[59]. Source data are provided with this paper.

### Code availability
All custom-written analysis scripts, the experiment script, and the stimuli used for the here presented results are freely available at https://doi.org/10.5281/zenodo.7941212[60].

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

## Acknowledgements

We thank Danielle E. Parrott for kindly serving as the ballet dancer in our video stimuli, and for giving informed consent for publishing her image in this article. We thank Gianpiero Monittola and Davide Tabarelli for their technical support in the MEG lab. We thank Christoph Huber-Huber for useful feedback on a previous version of the manuscript. I.E.J.V. is supported by an MSCA postdoctoral fellowship (101060807).

## Author contributions

Conceptualization: M.F.W. Methodology: I.E.J.V., M.F.W. Investigation: I.E.J.V., M.F.W. Visualization: I.E.J.V., M.F.W. Supervision: M.F.W. Writing—original draft: I.E.J.V. Writing—review & editing: I.E.J.V., M.F.W.

## Competing interests

The authors declare no competing interests.
