## [Peer Review File · Nature Communications]

Reviewer #1 (Remarks to the Author):

De Vries and Wurm aimed to understand how the brain supports perception of continuously unfolding dynamic stimuli, thereby requiring continuous prediction updating. They presented moving bodies and recorded MEG. They used a dynamic RSA approach to conclude that high level abstract (e.g., view invariant) features are predicted earlier in time, with lower level features predicted closer in time to the input. I found the manuscript interesting, especially the approach, but also did not follow some elements of the analysis, predictions and findings. I think these need to be clarified to generate a stronger sense of their likely significance.

Methods. To summarise what I understood, there are 14 videos. A 14x14 dissimilarity matrix is computed for these videos for each frame (/500) for each of 9 motion features (1: pixelwise similarity, 2: optical flow, 3:.. becoming more abstracted with each step). The authors then perform a temporal resampling step (that I didn't follow) and use the video information to predict the neural dissimilarity matrix across time, while including the other stimulus features as a nuisance regressor. This allowed the authors to ask about prediction of neural dissimilarity by stimulus dissimilarity and, therefore, the extent to which the particular neural representation was preceding the stimulus. However, I could not follow the temporal resampling step. "Temporal resampling" sounds unsettling given the nature of the question. I'm sure there is a logical explanation of it that does not alter interpretation but this needs to be clarified for the reader.

Prediction / interpretation. According to PC frameworks the higher (more abstract) levels pass information to lower (more specific) levels but I did not follow how this would be expected to result in the observed particular timecourse. The authors say that view-invariant body motion is associated with neural activity ~640-430 ms in advance, view-dependent ~300-150 ms in advance, and optical flow ~90 ms in advance. This is assuming a half second lag between the higher order visual prediction and the optic flow. This seems especially high no? Input will have reached the higher order regions in, say, 150 ms. I guess it can continue to be passed up the hierarchy before an updated prediction is passed down but would the authors hypothesise that a new prediction is only passed twice a second? Some theoretical unpacking of this result is required.

Relatedly, is there any account of cortical functioning that makes alternative predictions? The discussion of the theoretical possibilities and implications is fairly thin, which renders it harder to understand the significance of these findings. As the authors note, predictive activations have already been demonstrated in early visual regions (as well as the Blom PNAS paper, there is this one: <https://www.biorxiv.org/content/biorxiv/early/2022/06/29/2022.06.26.496535.full.pdf>). However, the findings in this latter paper are a bit different, such that there is evidence for preemptive processing in early regions but no further compensation with later regions. Why the apparent difference?

Findings. On the basis of the claims, I would have expected Figure 2 to show peaks for the different features at different predictive latencies (on the left of the graph). Such a peak is only clear for the view-dependent body motion. The optic flow direction also shows two peaks, with a sharp decline between them. How should the reader interpret this? Is this apparent high noise due to the new method or the underlying mechanism? Some more help to map the claims onto these panels would be appreciated.

Reviewer #2 (Remarks to the Author):

In this paper, de Vries and Wurm look at the timing of neural representations during dynamic movie watching, with the goal of disentangling predictive and lagged representations. They specifically look at representations of body posture and movement in videos of ballet dancing films. For describing such representations in dynamically unfolding videos, they develop an extension to the widely used representational similarity analysis (RSA) framework, which they term dynamic RSA (dRSA). In this framework, representational dissimilarity matrices (RDMs) obtained

from the neural data at each time point are compared to pre-specified model RDMs, which are constructed for every time point in the video. Comparing these two types of RDMs in turn yields a measure that can be used to describe the timing of neural responses relative to the video features: if a neural RDM at time t is best predicted by a predictor RDM extracted from a previous time point $t-X$, then the representation is lagged, whereas when the neural RDM is best predicted by a predictor RDM extracted from time point $t+X$, then the representation is predictive.

I really like the idea of the analysis, which is simple at first glance, but quite complex when looking more closely. I think the authors did a good job in addressing many of the issues that arise on the way, but I have a few more comments that I outline below. The data too look sensible, but also somewhat puzzling in parts. I'll elaborate on a few issues that may require further discussion below.

1.) It would be very nice to see the raw correlations with each RDM separately (e.g., the pixel RDM at each time point correlated with the neural RDM at each time point), without the massive semi-partial correlations that are applied from the outset by including all predictors in a single regression. I get that predictor correlation, and more specifically temporal correlations within predictors are a problem here, but it would be good if readers could judge for themselves how the data looks without too many complicated analysis steps performed "under the hood". I think this may also help to make the paper a bit easier to follow. Right now, everything is very condensed (too condensed for my taste), and I had to often jump back and forth between the results to the methods and vice versa to really understand what's going on. The fact that the first data I see are so highly processed already doesn't really help here.

2.) The procedure with the PCA regression which automatically select components that explain overall variance seems sensible, but it quite sophisticated and untransparent. I find it hard to appreciate whether this analysis indeed removed all temporal autocorrelations and influences of the other predictors. Would it be possible to supplement this with something simpler, which may not be the perfect analysis but is more tractable? For instance, one could simply try to take the "raw" predictor correlations, compute the peaks for each, and then perform partial correlation analysis where only the RDMs corresponding to the correlation peaks are controlled for? Likely, under this scenario, the correlations will not be perfectly orthogonal (as the beta weights hopefully are now), but it would be interesting to see if such a simple analysis could provide something similar? Another viable option would be to temporally bin the data more leniently (e.g., in bins of 100ms), which would allow you to do partial correlation analysis without an excessive number of predictors? I have a slight worry that pretty early predictive representations of movement may come about through complex interactions of the many unknown predictors in your regression (which are sometimes hard to trace). Or can you somehow otherwise reassure me that this can't be the case? The simulations are instructive, but it seems a bit unclear what happens when real neural data come into play that have all kinds of wild variance components.

3.) What I find quite surprising about the results is that the representations of movement are so clearly predictive and so early, and that, strikingly, there seems no "poststimulus" (or rather, lagged) representation of the movement. Is this not at odds with classical views of how vision works? Would this mean that accurate predictions just silence all lagged representations (and if it's so easy, why not do this for posture, too)? My naïve view would be that you must have a lagged representation of the same property (here: motion) in order to evaluate your predictions – and clearly, they're not always going to be 100% precise (specifically when you start making predictions for events that are hundreds of milliseconds in the future). I feel this lack of lagged representations is worth discussing.

4.) One could expect that early during the videos, and also when viewing them for the first time, predictive representations should be less strong, but that there would be stronger lagged representations (for instance due to stronger prediction errors due to more often inaccurate predictions). It would be interesting to see if they emerge more strongly over time across the experiment. Can something like this be seen in your data when you analyze snippets from different positions in the videos or in the experimental session?

5.) What's happening with optical flow direction in Panel 2A? Why this aggressive dip just after a

lag of zero? Is that an artifact of regressing out the other RDMs?

Reviewer #3 (Remarks to the Author):

This study by Vries & Wurm aims to provide a better understanding of how the brain dynamically predicts sensory inputs based on the rapidly changing external information. They introduce a new extension to conventional 'predictive processing' research by applying dynamic representational similarity analysis (dRSA) that uses temporally variable models to capture neural representations of predictions. This is a refreshing concept, as predictions have mainly been studied using static stimuli. They identify a hierarchical structure in predictive representations, where high-level abstract predictions precede lower-level features, while lower-level feature predictions appeared closer to stimulus onset. These results are impressive and very promising but it is not entirely clear whether the results provide sufficient evidence of predictions of future naturalistic stimuli continuously unfolding along the visual processing hierarchy. Also, several interpretations rest on descriptions of data features but are not accompanied by formal statistical tests. The paper is well written, if a bit dense and therefore lacking clarity on some methodological points in the main text.

Major issues

1. Do these results really provide evidence for predictive processing? What about other theories incorporating predictions but not prediction error, e.g. Lee & Mumford 2003, adaptive resonance (Grossberg)? See Aitchison & Lengyel 2017 for a discussion of Bayesian inference in the brain with and without predictive coding.
2. Do the results provide evidence for truly hierarchical, nested predictions, or could they also reflect independent streams of predictions at different levels? E.g., independent body motion and optical flow predictions?
3. There is not much background provided for why the authors chose the different type of stimulus/prediction models - maybe a short explanation of the different types of models in the Results part would help the reader navigate, and improve the flow of the paper.
4. On p. 4, the representation of body posture was found to positively lag – (according to the authors explanation, that means bottom-up processing), however, the authors argue that this lag is reflects top-down modulation, since it's early (70 ms). This seems a bit of a stretch, especially given that the authors set out in their introduction that they consider negative, but not positive, latencies a signature of predictions. Also, the explanation of "higher-level representation such as view-invariant body posture may have a larger processing latency compared to low-level features", is not really a clear explanation as view-invariant body motion is also a higher-level representation but this is accompanied by a negative lag.
5. On p. 4, the authors report temporally sustained activity for (pixelwise, optical flow magnitude and direction, view-invariant body posture). I'm not convinced of the interpretation of the sustained nature of the neural representations based on the information provided. Could these not be sustained simply because of autocorrelation in the stimulus, being continuous movies? It is not clear to me whether or not the correction for model autocorrelation takes this into account, since even if autocorrelation in the models is adjusted for, such autocorrelations may still exist in the *data*, right?
6. I did not understand the rationale behind the temporal resampling method. Can the authors explain this more clearly? Both the problem itself, and why resampling 3s windows provides a solution, were not very clear to me. Also, why the windows were offset but overlapping between videos was not clear. Doesn't a temporal offset destroy the alignment in temporal dynamics between videos? And is this analysis step critical, or would (qualitatively) similar results be obtained without it (i.e., simply taking the full 5s windows)?
7. Given that the 14 sequences were repeated many times, did the authors investigate whether

predictive effects were larger later in the experiment, when the sequences had become familiar?

8. Can the authors report the extent of shared variance between the models? They take shared variance into account in their analysis approach, but if covariance is too high the estimation of the models may become ill-posed.

Minor issues

1. The authors report on p. 2 that the temporal window preceding the actual input was around 640-430msec for view-invariant body motion and 300-150msec for view-dependent body motion, but these exact timings are not easily gleaned from the figures (Fig. 2a and Fig. 3a). It might be helpful to provide a finer grained x-axis for these figures.

2. Fig.2b., it could be a neat presentation of the source localised data to indicate the ROIs with the same colour scheme as for Fig.2a, that could highlight very clearly how the visual information is being processed on the visual hierarchy. Maybe the ROI figure (Fig. S3) could be added here?

3. For how many model entries (%) were eye position data missing?

4. The authors mention two different sets of cluster thresholds (strict and lenient); how were these combined?

Reviewer #1

De Vries and Wurm aimed to understand how the brain supports perception of continuously unfolding dynamic stimuli, thereby requiring continuous prediction updating. They presented moving bodies and recorded MEG. They used a dynamic RSA approach to conclude that high level abstract (e.g., view invariant) features are predicted earlier in time, with lower level features predicted closer in time to the input. I found the manuscript interesting, especially the approach, but also did not follow some elements of the analysis, predictions and findings. I think these need to be clarified to generate a stronger sense of their likely significance.

Methods. To summarise what I understood, there are 14 videos. A 14x14 dissimilarity matrix is computed for these videos for each frame (/500) for each of 9 motion features (1: pixelwise similarity, 2: optical flow, 3:.. becoming more abstracted with each step). The authors then perform a temporal resampling step (that I didn't follow) and use the video information to predict the neural dissimilarity matrix across time, while including the other stimulus features as a nuisance regressor. This allowed the authors to ask about prediction of neural dissimilarity by stimulus dissimilarity and, therefore, the extent to which the particular neural representation was preceding the stimulus. However, I could not follow the temporal resampling step. "Temporal resampling" sounds unsettling given the nature of the question. I'm sure there is a logical explanation of it that does not alter interpretation but this needs to be clarified for the reader.

We thank the reviewer for raising this important point on the temporal resampling step in our analysis pipeline (also voiced by reviewer 3, issue 6), and we agree that this step requires additional clarification. We now recognize that the term 'resampling' can be misleading, as it might suggest a shuffling/reordering of samples across time, which is not the case. Instead, a temporally intact 3-sec segment, or 'subsample', is randomly extracted from each of the 14 5-sec videos, after which these 14 new 3-sec segments are realigned. Crucially, while a different random 3-sec window was selected for each of the 14 stimuli, for a given stimulus the temporal alignment between neural signal and models remained intact (indicated by the vertical orange dotted lines in the new Fig. 7a). We now emphasize this important point the first time the subsampling step is mentioned in the caption of figure 1 on P25L813 and have expanded the explanation in the "*Temporal subsampling*" subsection of the methods, including a new figure to support that explanation (Fig. 7), and a new supplementary figure illustrating the complete ROI-based results without the temporal subsampling step (Fig. S3). Additionally, we have changed the term 'resampling' to 'subsampling' throughout the manuscript.

The updated "*Temporal subsampling*" section on P16L471 now reads:

"Our dRSA pipeline involved one additional important step to increase the generalizability of the results as well as the SNR. Imagine that at a given time point t_x , in all 14 videos the ballet dancer is in the middle of a figure and moving relatively slowly and predictably. At t_x prediction is possible far in advance, resulting in an early predictive dRSA peak. In contrast, if at t_y there is a transition between ballet figures in all 14 videos, prediction is not possible very far in advance, resulting in a later (or no) predictive dRSA peak. Last, if at t_z stimulus predictability is different for each of the 14 videos (likely the most occurring scenario), the predictive peak reflects an average thereof. In other words, stimulus-specific feature trajectories influence how far in advance a stimulus can be predicted, and thus the structure of the neural and model RDMs depends on the exact arbitrary pairwise alignment of the 14 ballet dancing sequences (and the alignment of event boundaries at different timescales⁵, such as onsets and offsets of individual ballet figures, subfigures, smaller motions, etc.). Consequently, the pattern across stimulus time in the 2-dimensional dRSA matrix is idiosyncratic and heterogeneous. If videos would have started a little earlier or later into the first ballet figure, the structure of RDMs, and therefore of the dRSA matrix, would look completely different at all time points. We are not interested in these specific 14 stimuli and their pairwise temporal alignment per se, but rather in dynamic prediction more generally. Ideally one would have an infinite number of stimuli, i.e., infinitely large RDMs.

To mimic an infinite number of stimuli, and thus minimize the idiosyncrasy of our dRSA results and increase SNR, we applied a temporal subsampling and realignment approach. Specifically, across 1000 iterations, a 3-sec segment was randomly extracted from each of the 14 5-sec stimuli (Fig. 7a; orange boxes). Next, these 14 'new' 3-sec segments were realigned to each

other, and RDMs were computed at each realigned time point t_R (Fig. 7b and c). Last, similarity between neural and model RDMs was computed for each realigned neural by model time, resulting in the 2-dimensional dRSA matrix. Crucially, within a single iteration, while a different random 3-sec window was selected for each of the 14 stimuli, for a given stimulus the temporal alignment between neural and model data remained intact (Fig. 7a; dotted vertical orange lines). This subsampling approach accomplishes a different pairwise alignment of the 14 stimuli on each iteration, thus creating completely new RDMs and temporal structure in the dRSA results. This is illustrated in Figure 7d: First, the dashed black line illustrates the results for V1 without the temporal subsampling step (taking the whole 5-sec videos; see Fig. S3 for the complete results). Second, the light gray lines illustrate the dRSA results on an example of 10 iterations. There is quite some variability in the shape of the curves, as well as the peak timings, which dependent on the arbitrary pairwise stimulus alignment on that given iteration. Last, the full black lines indicate the dRSA results after averaging the 1000 iterations. Note that this whole procedure is done on an individual subject level, and the averaged dRSA values are subsequently used for statistical analysis (see ‘*Statistical analysis*’). To summarize, this approach mimics having many more video stimuli that are differently aligned to each other, increasing SNR, and making the results become more generalizable (at least to other sets of ballet dancing videos).”

Prediction / interpretation. According to PC frameworks the higher (more abstract) levels pass information to lower (more specific) levels but I did not follow how this would be expected to result in the observed particular timecourse. The authors say that view-invariant body motion is associated with neural activity ~640-430 ms in advance, view-dependent ~300-150 ms in advance, and optical flow ~90 ms in advance. This is assuming a half second lag between the higher order visual prediction and the optic flow. This seems especially high no? Input will have reached the higher order regions in, say, 150 ms. I guess it can continue to be passed up the hierarchy before an updated prediction is passed down but would the authors hypothesise that a new prediction is only passed twice a second? Some theoretical unpacking of this result is required.

We thank the reviewer for making us aware that our previous explanation was too simplified, and we recognize that this may lead to a misinterpretation of the results. Two important aspects warrant more detailed consideration:

- 1) Most importantly, the three dRSA peaks do not evidence a serial cascade of events, and we cannot directly infer a causal relationship between the three different levels of prediction. In fact, we deem it unlikely that these predictions would purely follow such a strict serial cascade, i.e., that first view-invariant motion is predicted, which in turn enables later view-dependent motion prediction, which enables even later optical flow prediction. Presumably, the three levels of prediction even reflect partly independent streams, such that if high-level prediction (e.g., view-invariant motion) becomes impossible, low-level prediction (e.g., pixelwise motion) still happens to a certain extent (see also the response to the next issue). However, based on the hierarchical prediction literature, one would expect a causal relationship, such that high-level predictions modulate low-level predictions. How exactly this happens in naturalistic dynamic stimuli remains to be investigated, for instance by selectively perturbing high-level stimulus predictability. One expectation would be a shortening of the temporal forecast window for lower-level features (i.e., shifting the dRSA peak to be less predictive), or a decrease in the strength of prediction. We have now rewritten the section on hierarchical predictions to clarify this point starting on P3L76:

“...our results provide a first characterization of how neural prediction of future motion in naturalistic stimuli continuously unfolds **at several timescales** along the visual processing hierarchy. Specifically, peak latencies of predictive motion representations reflected the order along the processing hierarchy, such that view-invariant body motion was predicted earliest in time, followed by view-dependent body motion and optical flow vector direction (Fig. 3a; main effect of model in a post-hoc ANOVA with factors model and ROI; $F = 19.9$, $p = 8.3e-7$). **Note that this does not evidence a strict serial cascade of prediction, such that low-level prediction is only enabled by earlier higher-level prediction. Instead, some low-level prediction likely remains even without any high-level prediction, as is apparent from the exclusively low-level prediction of simple motion stimuli that do not have a complex naturalistic hierarchical structure^{6,7}.** The three levels of prediction might therefore partly

reflect independent streams. However, the temporal order of predictive representations observed here is in line with hierarchical Bayesian brain theories such as predictive coding **that** postulate that higher-level predictions act on, and therefore must precede, lower-level predictions⁸⁻¹⁰. Furthermore, such hierarchical prediction is also suggested by empirical findings from different sensory modalities¹¹⁻¹⁵, and complex contexts such as social perception and action observation^{16,17}. **In complex naturalistic stimuli as utilized here one would similarly expect a causal relationship amongst the partly independent prediction streams, such that high-level prediction affects low-level prediction.** It may be that predictions originating at a **higher, perceptually invariant** level, as best captured by view-invariant body motion, subsequently **modulate** more concrete predictions, to which eventually the sensory input can be compared. **Such causal relationship between different levels of prediction in naturalistic dynamic stimuli remains to be demonstrated in future research, for instance by selectively perturbing high-level stimulus predictability. Two expectations would be a shortening of the temporal forecast window for lower-level features (i.e., a less predictive dRSA peak), and a decrease in the strength of prediction.”**

- 2) Second, as explained in the response to the reviewer’s first issue, moment-by-moment feature predictability is determined by stimulus-specific feature trajectories and event boundaries. Combined with an arbitrary pairwise alignment between the 14 stimuli, this results in dRSA peak latencies that indicate when a stimulus feature is represented most strongly on average for this type of ballet dancing video. It does not mean this feature is always exactly and exclusively represented at that latency. Therefore, a peak latency of e.g., 500 msec does not indicate a prediction twice a second, but rather a continuous prediction, which for that particular feature is on average strongest 500 msec in advance. What we can therefore infer from our results is that on average high-level information is predicted half a second more in advance than low level information. While we hope this is now clear from our updated explanation in the “*Temporal subsampling*” section, we additionally clarify this at the first relevant point in the manuscript on P2L48:

“Further note that as moment-by-moment stimulus predictability is determined by stimulus-specific feature trajectories and event boundaries, the resulting dRSA peak reflects the average latency at which a feature is represented most strongly for these stimuli. It does not mean this feature is exclusively represented at that exact latency.”

Relatedly, is there any account of cortical functioning that makes alternative predictions? The discussion of the theoretical possibilities and implications is fairly thin, which renders it harder to understand the significance of these findings. As the authors note, predictive activations have already been demonstrated in early visual regions (as well as the Blom PNAS paper, there is this one: <https://www.biorxiv.org/content/biorxiv/early/2022/06/29/2022.06.26.496535.full.pdf>). However, the findings in this latter paper are a bit different, such that there is evidence for preemptive processing in early regions but no further compensation with later regions. Why the apparent difference?

We thank the reviewer for pointing us to this apparently contradictory literature, and with addressing the previous issue (incl. now mentioning the papers suggested by the reviewer) we hope to have also largely addressed this issue.

Most importantly, the two mentioned studies used simple stimuli, i.e., dots moving in straight lines (Johnson et al. 2022 BioRxiv), or apparent motion of circular sequences of wedges (Blom et al. 2020 PNAS). In contrast, we used complex naturalistic stimuli. One could expect predictive (concrete) representations of simple stimuli to be mainly found in early regions. In fact, our findings do not disagree with this: we find prediction for optical flow direction mainly in early regions, whereas prediction of e.g., view-dependent motion is more similar across early and late regions. What distinguishes the current study from earlier work is that 1) our dynamic RSA approach allows us to study both low- and higher-level prediction in more complex naturalistic stimuli, and 2) it enables us to distinguish these different levels of prediction in the same stimuli, thus revealing different temporal forecast windows of the brain depending on stimulus feature complexity or level of abstraction.

We now additionally mention this distinction with previous work in the last paragraph of the discussion on P6L185:

“The current results go beyond recent evidence for prediction of simple (apparently) moving stimuli in early brain regions^{6,7,18,19}, by showing that 1) dynamic RSA reveals prediction at both low and high levels of abstraction in complex naturalistic stimuli, and 2) by separating sources of variance it enables distinguishing these different levels in the same stimulus, thus revealing multiple feature-dependent temporal forecast windows in the brain.”

Findings. On the basis of the claims, I would have expected Figure 2 to show peaks for the different features at different predictive latencies (on the left of the graph). Such a peak is only clear for the view-dependent body motion. The optic flow direction also shows two peaks, with a sharp decline between them. How should the reader interpret this? Is this apparent high noise due to the new method or the underlying mechanism? Some more help to map the claims onto these panels would be appreciated.

The reviewer wonders why there is only a clear single peak for the view-dependent body motion, but not for optical flow direction, where two peaks are observed, nor for view-invariant body motion, where the predictive peak is less sharp (i.e., more spread out).

Concerning view-invariant body motion, as explained in the previous points it is hopefully now clear that we do not assume a single prediction exactly and exclusively at the peak latency. Additionally, we have now rewritten the section on representational spread to clarify these considerations further on P5L151:

“Interestingly, most neural representations were not exclusive to an exact latency, but spread over time, as quantified by comparing the amount of information spread in the neural signal surrounding a dRSA peak with the amount that can be expected based on model autocorrelation alone (i.e., representational spread or RS; Fig. 3b and c). Such spread could be caused by representations themselves being temporally sustained, or by variability in predictive latency across timepoints within a stimulus, across stimuli, or across subjects. One would expect more spread for higher-level predictions (i.e., of features changing at a longer timescale), and more precise timing (i.e., sharper peak) for low-level models with predictions close in time to the actual sensory input. This is indeed what we observe (Fig. 2a and 3c), i.e., while view-invariant body motion shows most representational spread on both sides of the peak, this is less so for view-dependent body motion. If in fact representational spread partly reflects a sustained representation, this might be explained by gradually, rather than abruptly, emerging predictions.”

Concerning the two peaks in the optical flow direction model: First note that our original *optical flow direction* model was constructed based on the complete optical flow vectors, which include a direction and a magnitude. However, our intention was to only use the vector direction (i.e., unit vector) but not magnitude, as the magnitude is already captured by the separate *optical flow magnitude* model. We now compute a pure *optical flow direction* model by taking the sine and cosine of the vector orientation in radians, which does not include magnitude information. This updated model shows a reduced decline. The sharp decline in the previous results may have therefore partly been caused by presence of magnitude information in both the optical flow magnitude and direction models. Concerning the remaining apparent two peaks, we have two reasons to believe that these indeed reflect both a predictive and a lagged representation and are not a confound of our new approach. Most importantly, if the two peaks and dip would be an artifact of our regression approach, one should see this in the simulations as well, which is not the case. We argue that particularly the analysis illustrated in figure 8c makes a strong case in this regard. Here the simulated data contains optical flow direction together with the strongest two models (pixelwise and optical flow magnitude). If optical flow direction is tested, while the other two are regressed out, this clearly results in a single rather than double peak. Second, the pattern looks very different across ROIs, which is more obvious in the below figure. This figure displays the same results as figure 2a without error shading and with only 3 ROIs. While the representation in V1 mainly has a predictive peak, the representation in LOTC mainly has a lagged peak, and aIPL has its peak at the latency at which the other ROIs have the ‘dip’. If the two peaks could be completely explained by a confound related to our stimuli, models, or regression approach, one would expect this to be similar across ROIs.

Reviewer #2

In this paper, de Vries and Wurm look at the timing of neural representations during dynamic movie watching, with the goal of disentangling predictive and lagged representations. They specifically look at representations of body posture and movement in videos of ballet dancing films. For describing such representations in dynamically unfolding videos, they develop an extension to the widely used representational similarity analysis (RSA) framework, which they term dynamic RSA (dRSA). In this framework, representational dissimilarity matrices (RDMs) obtained from the neural data at each time point are compared to pre-specified model RDMs, which are constructed for every time point in the video. Comparing these two types of RDMs in turn yields a measure that can be used to describe the timing of neural responses relative to the video features: if a neural RDM at time t is best predicted by a predictor RDM extracted from a previous time point $t-X$, then the representation is lagged, whereas when the neural RDM is best predicted by a predictor RDM extracted from time point $t+X$, then the representation is predictive.

I really like the idea of the analysis, which is simple at first glance, but quite complex when looking more closely. I think the authors did a good job in addressing many of the issues that arise on the way, but I have a few more comments that I outline below. The data too look sensible, but also somewhat puzzling in parts. I'll elaborate on a few issues that may require further discussion below.

1.) It would be very nice to see the raw correlations with each RDM separately (e.g., the pixel RDM at each time point correlated with the neural RDM at each time point), without the massive semi-partial correlations that are applied from the outset by including all predictors in a single regression. I get that predictor correlation, and more specifically temporal correlations within predictors are a problem here, but it would be good if readers could judge for themselves how the data looks without too many complicated analysis steps performed "under the hood". I think this may also help to make the paper a bit easier to follow. Right now, everything is very condensed (too condensed for my taste), and I had to often jump back and forth between the results to the methods and vice versa to really understand what's going on. The fact that the first data I see are so highly processed already doesn't really help here.

We appreciate the reviewer's suggestion to add the 'simpler' results that would be obtained if correlation coefficients instead of PCR regression weights are used as similarity measure and agree that the results of this more basic analysis should be available to all readers. We have therefore added a supplementary figure (Supplementary Fig. 2) with these results and a small additional clarification on P14L433:

"These problems are illustrated below for simulated data (see "Simulations", and particularly Fig. 8a). For completeness, the dRSA results using simple correlations are illustrated in Supplementary Figure 2. While these results (Supplementary Fig. 2) do reflect true shared variance between neural and model RDMs, caution is warranted with their interpretation as it is difficult to separate the contribution of different model RDMs. Note that both shared variance between models and temporal autocorrelation within models are not a problem with the dRSA approach per se, but rather inherent to naturalistic (dynamic) stimuli. We argue that our followed approach explained in the next paragraph offers a solution to separate the contribution of different model RDMs."

2.) The procedure with the PCA regression which automatically select components that explain overall variance seems sensible, but it quite sophisticated and untransparent. I find it hard to appreciate whether this analysis indeed removed all temporal autocorrelations and influences of the other predictors. Would it be possible to supplement this with something simpler, which may not be the perfect analysis but is more tractable? For instance, one could simply try to take the “raw” predictor correlations, compute the peaks for each, and then perform partial correlation analysis where only the RDMs corresponding to the correlation peaks are controlled for? Likely, under this scenario, the correlations will not be perfectly orthogonal (as the beta weights hopefully are now), but it would be interesting to see if such a simple analysis could provide something similar? Another viable option would be to temporally bin the data more leniently (e.g., in bins of 100ms), which would allow you to do partial correlation analysis without an excessive number of predictors? I have a slight worry that pretty early predictive representations of movement may come about through complex interactions of the many unknown predictors in your regression (which are sometimes hard to trace). Or can you somehow otherwise reassure me that this can’t be the case? The simulations are instructive, but it seems a bit unclear what happens when real neural data come into play that have all kinds of wild variance components.

We thank the reviewer for making us aware that we should have better supported our selection of the PCR approach rather than other approaches, which we now do. The reviewer suggests trying out a simpler, more tractable analysis than PCR, to confirm qualitative similarity, and to confirm that our results are not somehow caused by the relatively complex PCR approach. First, it is important to emphasize that the main goal of this analysis step is to maximally extract the variance in the neural signal explained by the model of interest, while ignoring the contribution of co-varying models. Using the simulations, we previously already explored several alternatives, amongst which those suggested by the reviewer. Importantly, these all performed worse at one end or the other: either at extracting variance explained by the model of interest, or at properly correcting for variance better explained by other models. We found PCR to be most effective in achieving both goals. On a sidenote, as the reviewer mentions, one important aspect of PCR is that PCA produces components that are orthogonal in their variance (i.e., no shared variance) therefore preventing problems of multicollinearity in the regression, something which is likely a problem for the suggested partial correlations (see below).

We reran the suggested alternatives and discuss the results here. Most importantly, the simulations indicate that both suggested approaches perform worse at attenuating effects of co-varying models, and that in some cases they produce quite extreme (spurious) negative values, likely due to multicollinearity or remaining cross-model correlations (see the new Fig. S4). We now briefly mention these control analyses, extra rationalization of using PCR, and refer to the new supplementary figure at P18L530:

“The goal of the simulations was to find a dRSA pipeline that was effective at testing only a single model of interest while regressing out other models with which the model of interest might share variance, and while regressing out the model of interest itself at distant (future or past) time points (i.e., to reduce effects of temporal autocorrelation). Principal component regression (see ‘*Dynamic representational similarity analysis*’) appeared **most** effective at achieving this goal. **Note that to prevent double-dipping**²⁰, we first fine-tuned the dRSA pipeline described above on the **simulated data, before subsequently applying** the exact same pipeline to the real neural data.”

And at P14L642:

“**Additionally, using the simulated data we explored several, perhaps more tractable, alternative approaches such as partial correlation to deal with the large amount of (covarying) predictor RDMs. Two example alternatives are illustrated in Figure S4. For the first approach based on partial correlation, peak latencies of the cross-model correlations were extracted from Figure 8a, after which a partial correlation that included the model of interest at a certain timepoint t_x , as well as all other models at their respective peak latencies in the cross-correlation with the model of interest (Fig. S4b). Since some cross-model correlations displayed two peaks (e.g., view-invariant body posture and motion; see Fig. 8a), a maximum of 2 peak latencies were selected for each to-partial-out model. The second approach based on partial correlation included besides the model of interest all other models after down sampling (250 msec; Fig. S4c). The simulations illustrated in Fig. S4 indicate that both alternatives perform worse at attenuating**

effects of co-varying models, and in some cases, they produce quite extreme (spurious) negative values, likely due to multicollinearity. Crucially, applying these approaches to the real data resulted in similar patterns as PCR (data not shown), indicating that the here presented results are not specific to the PCR approach.”

Additionally, below we present these analyses applied to the real data. Importantly, for both alternatives the pattern of results is very similar to that acquired with PCR, indicating that the presented results are not somehow caused by our specific PCR approach. Note that the slight overall reduction in similarity values observed in the figures below, likely have the same cause as the spurious negative values that are visible in the simulations (Fig. S4b and c) and that are almost completely absent when using the PCR approach (Fig. S4a). We think this might be due to multicollinearity not properly corrected for using the alternative approaches. Given the suboptimal results of these approaches applied to simulated data, we argue that PCR still is the best alternative.

- 1) The reviewer’s first suggestion is to take the “raw” predictor correlations (i.e., the cross-model correlations), compute the peak latency for each combination of models, and then perform partial correlation analysis where only the RDMs corresponding to the correlation peaks are controlled for. As one can see in Figure 8a, model autocorrelations show a broad pyramid shape, indicating that neighboring timepoints strongly correlate. Therefore, simply regressing out the models only at the peak cross-model correlation latency (which we in fact tried), or similarly as the reviewer suggests a partial correlation instead of regression, fail at properly attenuating variance from the other models. As one can see in Figure 8a, several cross-model correlations demonstrate multiple peaks. We therefore did not only include the other models at their relative peak latency but did so for a maximum of 2 peaks, which should improve correction for covarying models.

- 2) The reviewer’s second suggestion is to temporally bin the data more leniently (e.g., in bins of 100ms), which would allow for a partial correlation analysis without an excessive number of predictors. We also demonstrate this approach here and show that it was also less effective at achieving our goal. A few things to note. First, bins of 100 msec across latencies of -1.5 to 1.5 sec as we use in our PCR approach, would result in 30 predictors for 9 models, i.e., 270 predictors (with multicollinearity) to regress out. Having more predictors (270+1) than observations (91) is not possible for partial correlation due to rank deficiency of the regression matrix, similarly to regression. Hence the lowest sampling rate for the to-be-regressed out models is 250 msec, with a latency window limited to -1 to 1 sec, which is what we applied below.

3.) What I find quite surprising about the results is that the representations of movement are so clearly predictive and so early, and that, strikingly, there seems no “poststimulus” (or rather, lagged) representation of the movement. Is this not at odds with classical views of how vision works? Would this mean that accurate predictions just silence all lagged representations (and if it’s so easy, why not do this for posture, too)? My naïve view would be that you must have a lagged representation of the same property (here: motion) in order to evaluate your predictions – and clearly, they’re not always going to be 100% precise (specifically when you start making predictions for events that are hundreds of milliseconds in the future). I feel this lack of lagged representations is worth discussing.

We agree with the reviewer that this is an important aspect of our result that we failed to properly discuss previously, which we now do in the manuscript on P4L100:

“Interestingly, for low-level motion (i.e., optical flow direction) we did not only observe a predictive, but also a lagged post-stimulus representation. According to predictive processing theories, post-stimulus representations should reflect the difference between prediction and sensory input (i.e., unpredicted input or “prediction errors”) ^{21–25}. As such, it may be surprising that such post-stimulus representation is absent for high-level (view-dependent and -invariant) body motion. However, predictive processing theories hypothesize that accurate predictions effectively silence (or “explain away”) all sensory input ^{21–25}. Albeit speculative, it may thus be that due to the highly predictable nature of smooth biological motion as in the ballet stimuli used here, motion information is effectively silenced after the initial comparison at the lowest level, thus explaining the absence of lagged representation of high-level motion models. Such efficient silencing of redundant stimulus information might be particularly important in dynamic stimuli, as new potentially relevant input is continuously incoming. However, if accurate prediction has indeed silenced all sensory input in terms of high-level motion models, it is unclear why this does not hold for the posture models that clearly have lagged post-stimulus representations (see next section). Additionally, even for high-level motion information perfect prediction seems unlikely, and it is worth considering why dRSA might fail to pick up any remaining unpredicted information for high-level motion models. Note that dRSA reflects an average representation across stimulus time, stimuli, and subjects, rather than a single exclusive representation (latency). The little remaining unpredicted information may simply be too variable in latency to be picked up in the average representation. Also, the signal-to-noise ratio (SNR) might generally be lower for bottom-up representations, as the visual cortex receives a much denser network of feedback relative to feedforward projections ^{25,26}. In any case, these results do not provide conclusive evidence exclusively for predictive processing/coding theory (i.e., that accurate predictions silence all sensory input ²⁵) but leave the door open for related theories that do hypothesize lagged post-stimulus representations even after accurate predictions such as adaptive resonance ²⁷, or Bayesian inference without predictive coding ²⁸. Future research using dRSA should

selectively perturb stimulus predictability at different hierarchical levels as described above to arbitrate between these theories.”

4.) One could expect that early during the videos, and also when viewing them for the first time, predictive representations should be less strong, but that there would be stronger lagged representations (for instance due to stronger prediction errors due to more often inaccurate predictions). It would be interesting to see if they emerge more strongly over time across the experiment. Can something like this be seen in your data when you analyze snippets from different positions in the videos or in the experimental session?

We thank the reviewer for this interesting suggestion. First, we consider the improvement in predictions one might expect throughout the experiment. We would first like to clarify that it was very difficult for subjects to learn the ballet sequences, as the 14 different combinations of 4 ballet figures, selected from only 5 possible figures, made it hard to predict the next figure at any given time. E.g., if the first figure was a bow, the second could still be an arabesque, passé or jump, and was thus not predictable (see Table S1). This was confirmed by the fact that when anecdotally asked afterwards, subjects did not subjectively experience that they could predict the next ballet figure in the sequence. We therefore think these predictions are mainly enabled by prior knowledge on biological motion (and gravitation) that is already available to the subjects, rather than new stimulus knowledge acquired throughout the experiment. Having said that, we ran the dRSA analysis for the first and last 1/3 of trials and statistically compared these (see new Supplementary Figure 6). Interestingly, even if dRSA values are lower (due to lower SNR when averaging over less trials), the results of using only 1/3 of the trials are relatively comparable to the main results, indicating that dRSA works also with less data. Most importantly, we did not find any significant differences in the predictive representations, likely due to the reasons explained above. Interestingly, we did observe a reduction in the lagged representation of view-dependent body posture in the later trials, and a trend for a reduction in the lagged representation of optical flow direction in later trials, which would both be in line with a predictive processing account: i.e., lagged representations would only reflect prediction errors or new information and would thus be expected to reduce with more stimulus knowledge. However, only the effect in view-dependent body motion was significant, and only at our more lenient statistical threshold ($p < 0.05$), thus making this interpretation highly speculative. To test the reviewer's suggestion properly, it would be interesting in future research to e.g., compare expert observers to novices, or compare observers pre- and post-training of completely new stimulus material (i.e., not biological motion).

We have now added this analysis and explanation to the supplementary material, and refer to it on P9L269:

“With so many repetitions, one might expect stimulus familiarization throughout the experiment to improve prediction. However, in a post hoc dRSA analysis we did not observe robust differences between the first and last 1/3 of the experiment (see “Supplementary Discussion” and Supplementary Figure 6).”

Next, we consider the improvement in predictions one might expect within a video. For two reasons we believe this hypothesis and accompanying analysis are also not straightforward. First, as now better explained in the manuscript, the exact strength and latency of the predictive representations reflect an average over time points within a stimulus, across stimuli, and across subjects. While one would expect prediction to only ‘start up’ after the first few frames or so, after this initial phase prediction will mostly depend on the moment-by-moment predictability in each of the 14 videos. There is no reason to assume that the stimuli are e.g., easier to predict in the second vs. first half of the video. In fact, as the videos all start with relatively little/slow motion, which is relatively aligned across the 14 videos, one could expect stronger predictive representations at the very start of the video. For these reasons it is unclear what to expect from comparing the earlier and later parts of the videos, and the interpretation ambiguous. Second, to perform our dRSA analysis, one needs a certain minimum video length, and enough time for the temporal subsampling step. This makes it difficult to take the very early or late part of videos because the windows become too short. It is therefore inevitable to have a large overlap between the two time windows, making their comparison even less unambiguous.

Nevertheless, for the purpose of this response letter we tested this idea by running our exact dRSA analysis twice, once on the first 4 seconds of the videos, and once on the last 4 seconds of the videos

(i.e., by subsampling only between 0-4 sec and 1-5 sec, respectively). See the figures below, which do not show any robust differences.

Early (0-4 sec):

Late (1-5 sec):

5.) What's happening with optical flow direction in Panel 2A? Why this aggressive dip just after a lag of zero? Is that an artifact of regressing out the other RDMs?

First note that our original *optical flow direction* model was constructed based on the complete optical flow vectors, which include a direction and a magnitude. However, our intention was to only use the vector direction (i.e., unit vector) but not magnitude, as the magnitude is already captured by the separate *optical flow magnitude* model. We now compute a pure *optical flow direction* model by taking the sine and cosine of the vector orientation in radians, which does not include magnitude information. This updated model shows a reduced 'dip'. The 'aggressive' dip in the previous results may have therefore partly been caused by presence of magnitude information in both the optical flow magnitude and direction models. Concerning the remaining apparent two peaks, we have two reasons to believe that these indeed reflect both a predictive and a lagged representation and are not a confound of our new approach. Most importantly, if the two peaks and dip would be an artifact of our regression approach, one should see this in the simulations as well, which is not the case. We argue that particularly the analysis illustrated in figure 8c makes a strong case in this regard. Here the simulated

data contains optical flow direction together with the strongest two models (pixelwise and optical flow magnitude). If optical flow direction is tested, while the other two are regressed out, this clearly results in a single rather than double peak. Second, the pattern looks very different across ROIs, which is more obvious in the below figure. This figure displays the same results as figure 2a without error shading and with only 3 ROIs. While the representation in V1 mainly has a predictive peak, the representation in LOTC mainly has a lagged peak, and aIPL has its peak at the latency at which the other ROIs have the 'dip'. If the two peaks could be completely explained by a confound related to our stimuli, models, or regression approach, one would expect this to be similar across ROIs.

Reviewer #3

This study by Vries & Wurm aims to provide a better understanding of how the brain dynamically predicts sensory inputs based on the rapidly changing external information. They introduce a new extension to conventional 'predictive processing' research by applying dynamic representational similarity analysis (dRSA) that uses temporally variable models to capture neural representations of predictions. This is a refreshing concept, as predictions have mainly been studied using static stimuli. They identify a hierarchical structure in predictive representations, where high-level abstract predictions precede lower-level features, while lower-level feature predictions appeared closer to stimulus onset. These results are impressive and very promising but it is not entirely clear whether the results provide sufficient evidence of predictions of future naturalistic stimuli continuously unfolding along the visual processing hierarchy. Also, several interpretations rest on descriptions of data features but are not accompanied by formal statistical tests. The paper is well written, if a bit dense and therefore lacking clarity on some methodological points in the main text.

Major issues

1. Do these results really provide evidence for predictive processing? What about other theories incorporating predictions but not prediction error, e.g. Lee & Mumford 2003, adaptive resonance (Grossberg)? See Aitchison & Lengyel 2017 for a discussion of Bayesian inference in the brain with and without predictive coding.

We thank the reviewer for raising the important question of whether the here observed predictive representations provide evidence exclusively for the theory of predictive processing / coding, or whether they leave the door open for other related theories. The main difference between these theories is that predictive processing/coding specifically states that accurate predictions "explain away" sensory input, meaning that lagged post-stimulus representations only contain the unexplained input (i.e., prediction errors or new sensory input). In contrast, the other theories mentioned by the reviewer all propose that even in case of accurate predictions there should be a lagged post-stimulus representation. This is an important point, and while we believe there are some indications that our results specifically support predictive processing, we do agree with the reviewer that they do not conclusively exclude other related theories. First, we would like to point the reviewer to our new explanation on the absence of lagged post-stimulus representations for high-level motion, inspired by issue 3 of reviewer 2, on P4L100:

“Interestingly, for low-level motion (i.e., optical flow direction) we did not only observe a predictive, but also a lagged post-stimulus representation. According to predictive processing theories, post-stimulus representations should reflect the difference between prediction and sensory input (i.e., unpredicted input or “prediction errors”)^{21–25}. As such, it may be surprising that such post-stimulus representation is absent for high-level (view-dependent and -invariant) body motion. However, predictive processing theories hypothesize that accurate predictions effectively silence (or “explain away”) all sensory input^{21–25}. Albeit speculative, it may thus be that due to the highly predictable nature of smooth biological motion as in the ballet stimuli used here, motion information is effectively silenced after the initial comparison at the lowest level, thus explaining the absence of lagged representation of high-level motion models. Such efficient silencing of redundant stimulus information might be particularly important in dynamic stimuli, as new potentially relevant input is continuously incoming. However, if accurate prediction has indeed silenced all sensory input in terms of high-level motion models, it is unclear why this does not hold for the posture models that clearly have lagged post-stimulus representations (see next section). Additionally, even for high-level motion information perfect prediction seems unlikely, and it is worth considering why dRSA might fail to pick up any remaining unpredicted information for high-level motion models. Note that dRSA reflects an average representation across stimulus time, stimuli, and subjects, rather than a single exclusive representation (latency). The little remaining unpredicted information may simply be too variable in latency to be picked up in the average representation. Also, the signal-to-noise ratio (SNR) might generally be lower for bottom-up representations, as the visual cortex receives a much denser network of feedback relative to feedforward projections^{25,26}. In any case, these results do not provide conclusive evidence exclusively for predictive processing/coding theory (i.e., that accurate predictions silence all sensory input²⁵) but leave the door open for related theories that do hypothesize lagged post-stimulus representations even after accurate predictions such as adaptive resonance²⁷, or Bayesian inference without predictive coding²⁸. Future research using dRSA should selectively perturb stimulus predictability at different hierarchical levels as described above to arbitrate between these theories.”

Additionally, we now also mention those other theories in the final concluding paragraph on P7L188:

“In the current dataset it reveals hierarchical predictive representations of the future motion of an observed action, and an absence of post-stimulus motion representations at higher levels at which sensory input is possibly explained away by accurate prediction. While both these observations support predictive processing/coding theories^{8–10}, future research using dRSA is needed to exclude related theories such as adaptive resonance²⁷, or Bayesian inference without predictive coding²⁸.”

2. Do the results provide evidence for truly hierarchical, nested predictions, or could they also reflect independent streams of predictions at different levels? E.g., independent body motion and optical flow predictions?

We thank the reviewer for raising this important point, also related to issue 2 of reviewer 1. We agree with the reviewer that the three dRSA peaks observed here indeed by themselves do not provide evidence for truly hierarchical, nested predictions. That is, they do not evidence a serial cascade of predictions from high to low level, nor a causal relationship between the different levels. In fact, we deem it unlikely that these predictions would purely follow such a strict serial cascade, i.e., that first view-invariant motion is predicted, which in turn enables later view-dependent motion prediction, which enables even later optical flow prediction. Presumably, the three levels of prediction even reflect partly independent streams, as the reviewer suggests. Consequently, if high-level prediction (e.g., view-invariant motion) becomes impossible, low-level prediction (e.g., pixelwise motion) still happens to a certain extent. However, based on the hierarchical prediction literature, one would expect a causal relationship, such that high-level predictions at least modulate low-level predictions. How exactly this happens in naturalistic dynamic stimuli remains to be investigated, for instance by selectively perturbing high-level stimulus predictability. One expectation would be a shortening of the temporal forecast window for lower-level features (i.e., shifting the dRSA peak to be less predictive), or a decrease in the strength of prediction. We have now rewritten the section on hierarchical predictions to clarify this point starting on P3L76:

“...our results provide a first characterization of how neural prediction of future motion in naturalistic stimuli continuously unfolds **at several timescales** along the visual processing hierarchy. Specifically, peak latencies of predictive motion representations reflected the order along the processing hierarchy, such that view-invariant body motion was predicted earliest in time, followed by view-dependent body motion and optical flow vector direction (Fig. 3a; main effect of model in a post-hoc ANOVA with factors model and ROI; $F = 19.9$, $p = 8.3e-7$). **Note that this does not evidence a strict serial cascade of prediction, such that low-level prediction is only enabled by earlier higher-level prediction. Instead, some low-level prediction likely remains even without any high-level prediction, as is apparent from the exclusively low-level prediction of simple motion stimuli that do not have a complex naturalistic hierarchical structure^{6,7}. The three levels of prediction might therefore partly reflect independent streams. However, the temporal order of predictive representations observed here is in line with hierarchical Bayesian brain theories such as predictive coding that postulate that higher-level predictions act on, and therefore must precede, lower-level predictions⁸⁻¹⁰. Furthermore, such hierarchical prediction is also suggested by empirical findings from different sensory modalities¹¹⁻¹⁵, and complex contexts such as social perception and action observation^{16,17}. In complex naturalistic stimuli as utilized here one would similarly expect a causal relationship amongst the partly independent prediction streams, such that high-level prediction affects low-level prediction.** It may be that predictions originating at a conceptual level, as best captured by view-invariant body motion, subsequently **modulate** more concrete predictions, to which eventually the sensory input can be compared. **Such causal relationship between different levels of prediction in naturalistic dynamic stimuli remains to be demonstrated in future research, for instance by selectively perturbing high-level stimulus predictability. Two expectations would be a shortening of the temporal forecast window for lower-level features (i.e., a less predictive dRSA peak), and a decrease in the strength of prediction.**”

3. There is not much background provided for why the authors chose the different type of stimulus/prediction models - maybe a short explanation of the different types of models in the Results part would help the reader navigate, and improve the flow of the paper.

We thank the reviewer for this suggestion and agree that this will improve the flow of the paper. We have now added this explanation in the results section on P2L53:

“Here we applied dRSA to source-reconstructed **magnetoencephalography (MEG)** data of healthy human subjects observing dance videos that were modeled **at various levels of abstraction across the visual processing hierarchy, from low-level visual (i.e., pixelwise grayscale) to higher-level, perceptually more invariant (i.e., 3-dimensional view-dependent and -invariant body posture and motion), as to capture a comprehensive characterization of the dance sequences.**”

And expanded the previous explanation in the methods section on P9L274:

“We selected a total of 9 **stimulus-feature** models described in detail below (Fig. 5), plus a tenth subject-specific model capturing the eye position. **The reason for selecting these stimulus-feature models was to capture a comprehensive characterization of the dance sequences that encompasses several hierarchical levels of complexity/abstraction from low-level visual information (pixelwise) up to higher-level, perceptually more invariant information (view-invariant body posture and motion).**”

4. On p. 4, the representation of body posture was found to positively lag – (according to the authors explanation, that means bottom-up processing), however, the authors argue that this lag is reflects top-down modulation, since it’s early (70 ms). This seems a bit of a stretch, especially given that the authors set out in their introduction that they consider negative, but not positive, latencies a signature of predictions. Also, the explanation of “higher-level representation such as view-invariant body posture may have a larger processing latency compared to low-level features”, is not really a clear explanation as view-invariant body motion is also a higher-level representation but this is accompanied by a negative lag.

We thank the reviewer for making us aware of the confusion we may have created with this explanation. Most importantly, we argue that it would be impossible for view-invariant body posture (high-level) to be represented earlier (i.e., at 70 msec) than pixelwise grayscale (i.e., at 110 msec), if no top-down prediction or prior expectation would be involved at all. If the view-invariant body posture representation

would reflect mere bottom-up processing, how could one explain this representation to be earlier in time than a representation at a lower level along the visual processing hierarchy? To clarify the interpretation of this finding, we have rewritten the respective section on P5L127:

“We observed lagged representations of view-dependent (~200 msec; V3/4) and view-invariant (~70 msec; LOTC and aPL) body posture, which might suggest bottom-up activation. However, remember that a positive latency precludes inferring that a representation solely reflects bottom-up activation, as top-down expectations might very well modulate its veridicity¹² and processing latency¹³. In fact, we argue that the latency of the view-invariant body posture representation observed here is indeed modulated by top-down prediction. That is, in case of pure bottom-up processing along the visual hierarchy, one would expect a high-level representation such as view-invariant body posture in LOTC and aPL to have a larger processing latency compared to low-level representations – which is not what we observed. Instead, the dRSA peak latency for view-invariant body posture in LOTC (~70 msec) is shorter than the peak latency for the view-dependent body posture and pixelwise models (~200 and ~110 msec, resp.; Fig. 2a and 3a), suggesting that the observed neural representation of view-invariant body posture is at least partly shifted ahead in time by top-down prediction. This interpretation is in line with the observation that prediction shifts the neural representation of the position of simple (apparently) moving objects closer to real-time^{14,17}, and could explain the subjective experience of perceiving our dynamic environment in real-time rather than lagged^{17,37}. While this interpretation might seem intuitive regarding our conscious experience the world, a real-time position representation fails to explain how we are able to act promptly (e.g., catch a ball), as we would need some representation of the stimulus trajectory to reach the motor cortex clearly well ahead of real-time. Albeit speculative, it may therefore be that the close-to-real-time body posture representation explains our conscious experience, while in contrast the clearly predictive motion representations serve prompt behavior. Additionally, the exact prediction latencies as identified here, as well as those observed in previous studies, likely depend on stimulus and task characteristics. Future work is needed to identify which factors (e.g., speed, predictability, prior knowledge, attention) determine prediction latencies, and to clarify the apparent difference in representational latency between body posture (i.e., close to real-time) and body motion (i.e., clearly predictive).”

Additionally, we now clarify in the introduction why lagged dRSA latencies do not evidence exclusive bottom-up processing, and more generally what the dRSA latencies indicate, on P2L42:

“In case of pure bottom-up processing, one expects a lag between a time point-specific visual model state and the best-matching neural representation (i.e., the time needed for information to pass from retina to e.g., V1). In contrast, a negative lag should be observed for predictive neural representations, in which case neural representational content predicts the future model state. However, note that while a negative dRSA peak latency clearly evidences prediction, a positive latency precludes inferring that a representation is solely activated in a bottom-up manner. That is, even post-stimulus representations are likely to be modulated by top-down expectations³³, for instance by sharpening the representation²⁹, or by shortening the processing latency^{30,31}. Further note that as moment-by-moment stimulus predictability is determined by stimulus-specific feature trajectories and event boundaries, the resulting dRSA peak reflects the average latency at which a feature is represented most strongly for these stimuli. It does not mean this feature is exclusively represented at that exact latency.”

5. On p. 4, the authors report temporally sustained activity for (pixelwise, optical flow magnitude and direction, view-invariant body posture). I'm not convinced of the interpretation of the sustained nature of the neural representations based on the information provided. Could these not be sustained simply because of autocorrelation in the stimulus, being continuous movies? It is not clear to me whether or not the correction for model autocorrelation takes this into account, since even if autocorrelation in the models is adjusted for, such autocorrelations may still exist in the *data*, right?

To clarify, the model autocorrelation curves (Fig. 8a) capture all autocorrelation in the stimulus. This is exactly what we correct for in the representational spread analysis (Fig. 3b). The reviewer is correct that there is another source of autocorrelation, which is in the neural data. If we understand the reviewer correctly, the question is whether there is any neural autocorrelation unrelated to the stimuli that could explain the representational spread we observe. Importantly, the neural RDM is computed by first averaging ~36 trials of the exact same video, before computing pairwise dissimilarity between all 14

stimuli. By averaging over ~36 trials any neural autocorrelation that is not stimulus related is canceled out. Any remaining neural autocorrelation is thus related to the stimuli, and that is exactly what we are interested in. That is, if this stimulus-related neural autocorrelation is stronger than can be explained by the model autocorrelation, then it must be that the neural signal contains a spread-out representation of the model information. We now clarify this on P21L620:

“Note that there is another source of autocorrelation, i.e., in the neural data. However, one needs to remember that the neural RDM is computed by first averaging ~36 trials of the exact same video, before computing pairwise dissimilarity between all 14 stimuli. By averaging over ~36 trials any neural autocorrelation that is not stimulus related is canceled out. Any remaining neural autocorrelation is thus related to the stimuli, which is exactly what we are interested in. That is, if this stimulus-related neural autocorrelation is stronger than can be explained by the model autocorrelation, then it must be that the neural signal contains a spread-out representation of the model information.”

Having said that, we do realize now that a sustained neural representation is not the only possible explanation for the observed representational spread. This could also be caused by variability in the representation latency across stimulus time, across stimuli and across subjects. We have now rewritten the relevant results section, which now should also clarify this point, on P5L153:

“Such spread could be caused by representations themselves being temporally sustained, or by variability in predictive latency across timepoints within a stimulus, across stimuli, or across subjects.”

And on P6L159:

“If in fact representational spread partly reflects a sustained representation, this might be explained by gradually, rather than abruptly, emerging predictions.

For lagged representations, this could indicate that representations were not immediately overwritten by new incoming visual input but remained partly activated.”

6. I did not understand the rationale behind the temporal resampling method. Can the authors explain this more clearly? Both the problem itself, and why resampling 3s windows provides a solution, were not very clear to me. Also, why the windows were offset but overlapping between videos was not clear. Doesn't a temporal offset destroy the alignment in temporal dynamics between videos? And is this analysis step critical, or would (qualitatively) similar results be obtained without it (i.e., simply taking the full 5s windows)?

We thank the reviewer for raising this important point on the temporal resampling step in our analysis pipeline (also voiced by reviewer 1, issue 1), and we agree that this step requires additional clarification. We now recognize that the term ‘resampling’ can be misleading, as it might suggest a shuffling/reordering of samples across time, which is not the case. Instead, a temporally intact 3-sec segment, or ‘subsample’, is randomly extracted from each of the 14 5-sec videos, after which these 14 new 3-sec segments are realigned. Crucially, while a different random 3-sec window was selected for each of the 14 stimuli, for a given stimulus the temporal alignment between neural signal and models remained intact (indicated by the vertical orange dotted lines in the new Fig. 7b). We now emphasize this important point the first time the subsampling step is mentioned in the caption of figure 1 on P25L813 and have expanded the explanation in the “*Temporal subsampling*” subsection of the methods, including a new figure to support that explanation (Fig. 7), and a new supplementary figure illustrating the complete ROI-based results without the temporal subsampling step (i.e., simply taking the full 5-sec windows; Fig. S3). Additionally, we have changed the term ‘resampling’ to ‘subsampling’ throughout the manuscript.

The updated “*Temporal subsampling*” section on P16L471 now reads:

“Our dRSA pipeline involved one additional important step to increase the generalizability of the results as well as the SNR. Imagine that at a given time point t_x , in all 14 videos the ballet dancer is in the middle of a figure and moving relatively slowly and predictably. At t_x prediction is possible far in advance, resulting in an early predictive dRSA peak. In contrast, if at t_y there

is a transition between ballet figures in all 14 videos, prediction is not possible very far in advance, resulting in a later (or no) predictive dRSA peak. Last, if at t_z stimulus predictability is different for each of the 14 videos (likely the most occurring scenario), the predictive peak reflects an average thereof. In other words, stimulus-specific feature trajectories influence how far in advance a stimulus can be predicted, and thus the structure of the neural and model RDMS depends on the exact arbitrary pairwise alignment of the 14 ballet dancing sequences (and the alignment of event boundaries at different timescales⁵, such as onsets and offsets of individual ballet figures, subfigures, smaller motions, etc.). Consequently, the pattern across stimulus time in the 2-dimensional dRSA matrix is idiosyncratic and heterogeneous. If videos would have started a little earlier or later into the first ballet figure, the structure of RDMS, and therefore of the dRSA matrix, would look completely different at all time points. We are not interested in these specific 14 stimuli and their pairwise temporal alignment per se, but rather in dynamic prediction more generally. Ideally one would have an infinite number of stimuli, i.e., infinitely large RDMS.

To mimic an infinite number of stimuli, and thus minimize the idiosyncrasy of our dRSA results and increase SNR, we applied a temporal subsampling and realignment approach. Specifically, across 1000 iterations, a 3-sec segment was randomly extracted from each of the 14 5-sec stimuli (Fig. 7a; orange boxes). Next, these 14 'new' 3-sec segments were realigned to each other, and RDMS were computed at each realigned time point t_R (Fig. 7b and c). Last, similarity between neural and model RDMS was computed for each realigned neural by model time, resulting in the 2-dimensional dRSA matrix. Crucially, within a single iteration, while a different random 3-sec window was selected for each of the 14 stimuli, for a given stimulus the temporal alignment between neural and model data remained intact (Fig. 7a; dotted vertical orange lines). This subsampling approach accomplishes a different pairwise alignment of the 14 stimuli on each iteration, thus creating completely new RDMS and temporal structure in the dRSA results. This is illustrated in Figure 7d: First, the dashed black line illustrates the results for V1 without the temporal subsampling step (taking the whole 5-sec videos; see Fig. S3 for the complete results). Second, the light gray lines illustrate the dRSA results on an example of 10 iterations. There is quite some variability in the shape of the curves, as well as the peak timings, which dependent on the arbitrary pairwise stimulus alignment on that given iteration. Last, the full black lines indicate the dRSA results after averaging the 1000 iterations. Note that this whole procedure is done on an individual subject level, and the averaged dRSA values are subsequently used for statistical analysis (see '*Statistical analysis*'). To summarize, this approach mimics having many more video stimuli that are differently aligned to each other, increasing SNR, and making the results become more generalizable (at least to other sets of ballet dancing videos)."

7. Given that the 14 sequences were repeated many times, did the authors investigate whether predictive effects were larger later in the experiment, when the sequences had become familiar?

We thank the reviewer for this interesting suggestion. We would first like to clarify that it was very difficult for subjects to learn the ballet sequences, as the 14 different combinations of 4 ballet figures, selected from only 5 possible figures, made it hard to predict the next figure at any given time. E.g., if the first figure was a bow, the second could still be an arabesque, passé or jump, and was thus not predictable (see Table S1). This was confirmed by the fact that when anecdotally asked afterwards, subjects did not subjectively experience that they could predict the next ballet figure in the sequence. We therefore think these predictions are mainly enabled by prior knowledge on biological motion (and gravitation) that is already available to the subjects, rather than new stimulus knowledge acquired throughout the experiment. Having said that, we ran the dRSA analysis for the first and last 1/3 of trials and statistically compared these (see new Supplementary Figure 6). Interestingly, even if dRSA values are lower (due to lower SNR when averaging over less trials), the results of using only 1/3 of the trials are relatively comparable to the main results, indicating that dRSA works also with less data. Most importantly, we did not find any significant differences in the predictive representations, likely due to the reasons explained above. Interestingly, we did observe a reduction in the lagged representation of view-dependent body posture in the later trials, and a trend for a reduction in the lagged representation of optical flow direction in later trials, which would both be in line with a predictive processing account: i.e., lagged representations would only reflect prediction errors or new information and would thus be expected to reduce with more stimulus knowledge. However, only the effect in view-dependent body motion was significant, and only at our more lenient statistical threshold ($p < 0.05$), thus making this

interpretation highly speculative. To test the reviewer's suggestion properly, it would be interesting in future research to e.g., compare expert observers to novices, or compare observers pre- and post-training of completely new stimulus material (i.e., not biological motion).

We have now added this analysis and explanation to the supplementary material, and refer to it on P9L269:

“With so many repetitions, one might expect stimulus familiarization throughout the experiment to improve prediction. However, in a post hoc dRSA analysis we did not observe robust differences between the first and last 1/3 of the experiment (see “Supplementary Discussion” and Supplementary Figure 6).”

8. Can the authors report the extent of shared variance between the models? They take shared variance into account in their analysis approach, but if covariance is too high the estimation of the models may become ill-posed.

First, the shared variance between models is already visible in Figure 8a, i.e., variance is correlation squared. For visibility we have now added a supplementary figure (Fig. S5) in which we show shared variance as correlation squared and multiplied by 100%, and refer to this figure on P18L538:

“...see also Supplementary Figure 5 for shared variance, i.e., correlation squared)”

Second, it is important to note that the here used regression approach, i.e., principal component regression, removes any covariance between models with the PCA step. Since all predictors are entered into a PCA before the regression, the resulting components that are used for the regression are inherently independent/orthogonal (i.e., no covariance). This is a big advantage of the PCR approach (see also our reply to issue 2 of reviewer 2 in which we explore alternatives to our regression approach that do not have this advantage). Last, note that the result of applying PCR to the simulated data, as illustrated in Figure 8b, confirm that nothing strange is going on (e.g., such as arbitrary switches in the assignment of beta weights to models that one might expect in an ill-posed problem).

To emphasize this advantage of PCR we have changed the sentence on P15L461 from:

“...hence reducing the risk of multicollinearity.”
to
“...hence **preventing** multicollinearity.”

Minor issues

1. The authors report on p. 2 that the temporal window preceding the actual input was around 640-430msec for view-invariant body motion and 300-150msec for view-dependent body motion, but these exact timings are not easily gleaned from the figures (Fig. 2a and Fig. 3a). It might be helpful to provide a finer grained x-axis for these figures.

In figure 2a and 3c we have increased the number of x-ticks to steps of 0.25 sec, with a label at each 0.5 sec. In figure 3a we have increased the number of x-ticks to steps of 0.1 sec. Additionally, we have increased the length of all x-ticks. We feel that these changes improve readability enough, without cluttering the figures with labels.

2. Fig.2b., it could be a neat presentation of the source localised data to indicate the ROIs with the same colour scheme as for Fig.2a, that could highlight very clearly how the visual information is being processed on the visual hierarchy. Maybe the ROI figure (Fig. S3) could be added here?

We thank the reviewer for this nice idea. Unfortunately, after some tries, we realized that overlaying the cortical surface plots with the ROI information (e.g., ROI outlines in the respective colors used in 2a) doesn't work well as the cortical surface plots become too clustered with information in different colors. However, we do agree it improves readability if readers have easier access to the ROI figure (former Fig. S3), which is now placed in the methods section of the main manuscript as Figure 6. Additionally, we now refer to this figure in the caption of figure 2b.

3. For how many model entries (%) were eye position data missing?

We have now added this information on P11L322:

“The raw eye tracker signals were first downsampled to 100 Hz, after which trials in which the eye tracker lost the eye for more than 10% of the samples were removed (**a total of 1.0% of trials was missing; min–max across subjects = 0–5.9%**), and the remaining missing samples were interpolated (e.g., due to blinks; **a total of 0.3% of samples; min–max across subjects = 0–2.1%**).”

4. The authors mention two different sets of cluster thresholds (strict and lenient); how were these combined?

This was already previously explained in the caption of figure 2:

“Horizontal bars indicate significant beta weights ($p < 0.01$ for single time samples), cluster-corrected for multiple comparisons across time ($p < 0.05$ for cluster-size), with colors matching the respective ROI line plot. Black or white horizontal bars inside colored bars indicate significance at a stricter statistical threshold (i.e., $p < 0.001$ for single samples and $p < 0.01$ for cluster-size).”

For clarity, we now also mention this in the relevant methods section on P17L524:

“In the main results (Fig. 2a), lenient significant intervals are indicated by thick horizontal bars with colors matching the respective ROI line plots, while strict significant intervals are indicated with thin black or white horizontal bars inside the colored bars.”

Note that the reason for using black OR white for the strict significant intervals, is purely for visibility, i.e., the black horizontal bars are more visible on top of the tick horizontal bars with light colors, while the white thin horizontal bars are more visible on top of the thick horizontal bars with dark colors.

References

1. Kiesel, A., Miller, J., Jolicœur, P. & Brisson, B. Measurement of ERP latency differences: A comparison of single-participant and jackknife-based scoring methods. *Psychophysiology* **45**, 250–274 (2008).
2. Smulders, F. T. Y. Simplifying jackknifing of ERPs and getting more out of it: Retrieving estimates of participants' latencies. *Psychophysiology* **47**, 387–392 (2010).
3. Kiesel, A., Miller, J., Jolicœur, P. & Brisson, B. Measurement of ERP latency differences: A comparison of single-participant and jackknife-based scoring methods. *Psychophysiology* **45**, 250–274 (2008).
4. Smulders, F. T. Y. Simplifying jackknifing of ERPs and getting more out of it: Retrieving estimates of participants' latencies. *Psychophysiology* **47**, 387–392 (2010).
5. Baldassano, C. *et al.* Discovering Event Structure in Continuous Narrative Perception and Memory. *Neuron* **95**, 709–721.e5 (2017).
6. Blom, T., Feuerriegel, D., Johnson, P., Bode, S. & Hogendoorn, H. Predictions drive neural representations of visual events ahead of incoming sensory information. *Proc Natl Acad Sci U S A* **117**, 7510–7515 (2020).
7. Johnson, P. A. *et al.* Position representations of moving objects align with real-time position in the early visual response. *Elife* **12**, (2023).
8. Clark, A. Whatever next? Predictive brains, situated agents, and the future of cognitive science. *Behavioral and Brain Sciences* **36**, 181–204 (2013).
9. Rao, R. P. N. & Ballard, D. H. Predictive coding in the visual cortex: A functional interpretation of some extra-classical receptive-field effects. *Nat Neurosci* **2**, 79–87 (1999).

10. Bastos, A. M. *et al.* Canonical Microcircuits for Predictive Coding. *Neuron* **76**, 695–711 (2012).
11. Heilbron, M., Armeni, K., Schoffelen, J. M., Hagoort, P. & De Lange, F. P. A hierarchy of linguistic predictions during natural language comprehension. *Proc Natl Acad Sci U S A* **119**, 1–12 (2022).
12. de Lange, F. P., Heilbron, M. & Kok, P. How Do Expectations Shape Perception? *Trends Cogn Sci* **22**, 764–779 (2018).
13. Gayet, S. & Peelen, M. V. Preparatory attention incorporates contextual expectations. *Current Biology* **32**, 687–692.e6 (2022).
14. Wacongne, C. *et al.* Evidence for a hierarchy of predictions and prediction errors in human cortex. *Proc Natl Acad Sci U S A* **108**, 20754–20759 (2011).
15. Kok, P. & De Lange, F. P. Shape perception simultaneously up- and downregulates neural activity in the primary visual cortex. *Current Biology* **24**, 1531–1535 (2014).
16. Bach, P. & Schenke, K. C. Predictive social perception: Towards a unifying framework from action observation to person knowledge. *Soc Personal Psychol Compass* **11**, 1–17 (2017).
17. Kilner, J. M., Friston, K. J. & Frith, C. D. Predictive coding: An account of the mirror neuron system. *Cogn Process* **8**, 159–166 (2007).
18. Ekman, M., Kok, P. & De Lange, F. P. Time-compressed preplay of anticipated events in human primary visual cortex. *Nat Commun* **8**, 1–9 (2017).
19. Ekman, M., Kusch, S. & de Lange, F. P. Successor-like representation guides the prediction of future events in human visual cortex and hippocampus. *bioRxiv* (2022) doi:<https://doi.org/10.1101/2022.03.23.485480>.
20. Kriegeskorte, N., Simmons, W. K., Bellgowan, P. S. & Baker, C. I. Circular analysis in systems neuroscience: The dangers of double dipping. *Nat Neurosci* **12**, 535–540 (2009).
21. Clark, A. Whatever next? Predictive brains, situated agents, and the future of cognitive science. *Behavioral and Brain Sciences* **36**, 181–204 (2013).
22. Rao, R. P. N. & Ballard, D. H. Predictive coding in the visual cortex: A functional interpretation of some extra-classical receptive-field effects. *Nat Neurosci* **2**, 79–87 (1999).
23. Summerfield, C. & De Lange, F. P. Expectation in perceptual decision making: Neural and computational mechanisms. *Nat Rev Neurosci* **15**, 745–756 (2014).
24. Friston, K. The free-energy principle: A unified brain theory? *Nat Rev Neurosci* **11**, 127–138 (2010).
25. Walsh, K. S., McGovern, D. P., Clark, A. & O’Connell, R. G. Evaluating the neurophysiological evidence for predictive processing as a model of perception. *Annals of the New York Academy of Sciences* vol. 1464 242–268 Preprint at <https://doi.org/10.1111/nyas.14321> (2020).
26. Van Essen, D. C. & Maunsell, J. H. R. Hierarchical organization and functional streams in the visual cortex. *Trends Neurosci* **6**, 370–375 (1983).
27. Lee, T. S. & Mumford, D. Hierarchical Bayesian inference in the visual cortex. *Journal of the Optical Society of America A* **20**, 1434 (2003).
28. Aitchison, L. & Lengyel, M. With or without you: predictive coding and Bayesian inference in the brain. *Current Opinion in Neurobiology* vol. 46 219–227 Preprint at <https://doi.org/10.1016/j.conb.2017.08.010> (2017).
29. Kok, P., Jehee, J. F. M. & de Lange, F. P. Less Is More: Expectation Sharpens Representations in the Primary Visual Cortex. *Neuron* **75**, 265–270 (2012).

30. Pinto, Y., van Gaal, S., de Lange, F. P., Lamme, V. A. F. & Seth, A. K. Expectations accelerate entry of visual stimuli into awareness. *J Vis* **15**, (2015).
31. Hogendoorn, H. & Burkitt, A. N. Predictive coding of visual object position ahead of moving objects revealed by time-resolved EEG decoding. *Neuroimage* **171**, 55–61 (2018).
32. Hogendoorn, H. Perception in real-time: predicting the present, reconstructing the past. *Trends Cogn Sci* **26**, 128–141 (2022).
33. de Lange, F. P., Heilbron, M. & Kok, P. How Do Expectations Shape Perception? *Trends Cogn Sci* **22**, 764–779 (2018).

Reviewer #1 (Remarks to the Author):

The authors have addressed my concerns and improved their manuscript accordingly. I believe it is now suitable for publication.

Clare Press

Reviewer #2 (Remarks to the Author):

The authors did a very good job in thoroughly addressing my comments. The additional analyses and discussion improved the paper and I have no further concerns.

Reviewer #3 (Remarks to the Author):

The revisions have substantially improved the manuscript, and I have no further concerns. I remain enthusiastic about the methods and the findings.